# Primary somatosensory cortex bidirectionally modulates sensory gain and nociceptive behavior in a layer-specific manner

Katharina Ziegler [1,5], Ross Folkard[1,5], Antonio J. Gonzalez[1,5], Jan Burghardt[1], Sailaja Antharvedi-Goda[1], Jesus Martin-Cortecero [1], Emilio Isaías-Camacho [1], Sanjeev Kaushalya[2], Linette Liqi Tan[2], Thomas Kuner[3], Claudio Acuna[4], Rohini Kuner [2], Rebecca Audrey Mease [1,6] ✉ & Alexander Groh[1,6] ✉

The primary somatosensory cortex (S1) is a hub for body sensation of both innocuous and noxious signals, yet its role in somatosensation versus pain is debated. Despite known contributions of S1 to sensory gain modulation, its causal involvement in subjective sensory experiences remains elusive. Here, in mouse S1, we reveal the involvement of cortical output neurons in layers 5 (L5) and 6 (L6) in the perception of innocuous and noxious somatosensory signals. We find that L6 activation can drive aversive hypersensitivity and spontaneous nocifensive behavior. Linking behavior to neuronal mechanisms, we find that L6 enhances thalamic somatosensory responses, and in parallel, strongly suppresses L5 neurons. Directly suppressing L5 reproduced the pronociceptive phenotype induced by L6 activation, suggesting an anti-nociceptive function for L5 output. Indeed, L5 activation reduced sensory sensitivity and reversed inflammatory allodynia. Together, these findings reveal a layer-specific and bidirectional role for S1 in modulating subjective sensory experiences.

Flexible modulation of sensory gain is a canonical thalamocortical circuit function, serving to sharpen sensory information streams en route to the cortex[1–3] and thereby adaptively drive perception and behavior[4]. Cortical output pathways from layers 5 and 6 (L5 and L6) are in key positions to control sensory gain via their influence on diverse subcortical structures[5,6]. The emerging consensus is that L6 corticothalamic (L6-CT) neurons of primary sensory cortices control sensory gain by both recruiting cortical inhibitory neurons[7–9] and modulating multiple cellular and circuit mechanisms in the thalamus[1,10–14]. These L6-CT functions have been evaluated from the standpoint of gain control as beneficial to sensory processing, attention, and perception[1,10,11,13,15]. However, such powerful sensitivity tuning could give rise to a maladaptive mismatch between cortical gain and the sensory environment, for example, by promoting hypersensitivity to peripheral sensory stimuli. Such a potentially detrimental facet of L6-CT gain modulation could be particularly pertinent to sensory circuits involved in pain, as these circuits link sensory and affective functions to generate both healthy and pathological pain perceptions[16–18].

[1]Medical Biophysics, Institute for Physiology and Pathophysiology, Heidelberg University, Heidelberg, Germany. [2]Department of Molecular Pharmacology, Institute for Pharmacology, Heidelberg University, Heidelberg, Germany. [3]Institute for Anatomy and Cell Biology, Heidelberg University, Heidelberg, Germany. [4]Chica and Heinz Schaller Research Group, Institute for Anatomy and Cell Biology, Heidelberg University, Heidelberg, Germany. [5]These authors contributed equally: Katharina Ziegler, Ross Folkard, Antonio J Gonzalez. [6]These authors jointly supervised this work: Rebecca A Mease, Alexander Groh. ✉e-mail: beckin@gmail.com; groh@uni-heidelberg.de

Primary somatosensory cortex (S1) receives both innocuous and noxious signals via thalamocortical (TC) inputs from the ventral posterolateral (VPL) thalamus, and is, therefore, a primary junction at which body sensations could be amplified into pain or, conversely, painful signals could be suppressed[17,19]. In turn, S1's output is broadcast to thalamic nuclei via the L6-CT pathway and to several subcortical targets via the L5 pathway[6,20]. Thus, S1 could potentially modulate ascending pain signals at various subcortical points, including VPL, which is tightly linked to different pain states[21–28]. Indeed, tantalizing yet heterogeneous reports that broad manipulations of S1 alter sensory and affective pain responses[29–33] have challenged the view that S1 generates strictly sensory-discriminative aspects of pain[16,34–37]. However, the prevalent use of bulk manipulations and lesions of S1[38,39] has led to contrasting views on the role of S1 in pain processing[40,41]. Therefore, dissecting the cell-type-specific contributions of distinct S1 output layers to pain perception is required to disambiguate potential opposing effects which may underlie these controversies.

We hypothesized a link between S1 gain modulation and the subjective dimensions of sensory experiences, which may contribute to the generation of painful sensations. To test this hypothesis, we probed how L6-CT neurons in S1 modulate sensitivity to innocuous and noxious stimuli in mice, using sensory and affective behavioral read-outs of pain, targeted optogenetics, and electrophysiology in the VPL-S1 TC system.

Here, we report that L6-CT−but not L5−activation triggers mechanical hypersensitivity and negative affect (i.e., aversion) in behaving mice, suggesting that L6-CT gain amplification can contribute to functionally adverse outcomes in sensory processing. Probing the neural basis of this effect with in vivo electrophysiology demonstrated that L6-CT activation enhances TC signaling at the level of VPL and superficial layers of S1. In parallel, L6-CT stimulation largely silences L5 neurons suggesting that the pronociceptive effect of L6-CT results from both enhancements of thalamocortical gain and simultaneous suppression of L5. Indeed, direct optogenetic inhibition of L5 elicited mechanical hypersensitivity, negative affect (i.e., aversion), and excitation of L6. Conversely, L5 activation had antinociceptive effects, namely a decrease in mechanical sensitivity in naive animals, and in animals with pharmacologically-induced inflammatory allodynia, affective L5-stimulation-seeking behavior, and a complete reversal of mechanical hypersensitivity. Together, these findings reveal bidirectional control over somatosensation by two discrete S1 cortical output pathways, suggesting future avenues for treating pain-related and sensory processing disorders.

## Results

The Ntsr1-Cre mouse line is a useful model to study L6-CT pathways in different sensory cortices, including S1 cortex[7,8,10,13,14,20,42]. To assess the role of L6-CT neurons in modulating sensory gain and nociception, we expressed channelrhodopsin-2-EYFP (ChR2) or EGFP in L6-CT neurons of the right S1 hindlimb cortex (S1HL) of Ntsr1-Cre mice via stereotaxic injections of AAV-DIO-ChR2-EYFP (L6-ChR2 mice) or AAV-DIO-EGFP (L6-EGFP mice). Transgene expression in L6-ChR2 and L6-EGFP mice was restricted to L6 neurons in S1HL and their corticothalamic axons/terminals (Fig. 1a; L6-EGFP Supplementary Fig. 1b). Extracellular silicon-probe recordings in anesthetized L6-ChR2 mice validated optotagging of L6-CT neurons in S1HL (Fig. 1b and Supplementary Fig. 2b, c), and demonstrated persistent drive of L6-CT spiking activity during 5 s laser stimuli (Fig. 1c) in a laser intensity-dependent manner (Fig. 1d).

### Optogenetically-evoked L6-CT activity in the S1HL cortex elicits spontaneous nocifensive behavior

We asked whether L6-CT activation in the S1HL cortex has any measurable effect on spontaneous behavior in the absence of peripheral sensory stimulation. In awake, freely moving animals implanted with a

fiber optic above S1HL (Fig.1e), optogenetic activation of L6-CT neurons elicited vigorous lifting and shaking of the contralateral hindlimb in a laser intensity-dependent manner (Fig. 1f and Supplementary video 1). At maximum intensity (318 mW/mm$^2$), the majority of animals showed paw lifting and limb shaking (9/11 and 6/11, respectively), while EGFP-expressing control animals did not show any responses to the laser. The manifestation of these vigorous nocifensive responses, specifically to the contralateral hindlimb, suggests that L6-CT activation in S1HL evokes a nociceptive experience originating from the hindlimb, even in the absence of peripheral sensory stimulation.

### Optogenetically-evoked L6-CT activity in the S1HL cortex induces hypersensitivity, exacerbates inflammatory allodynia, and induces aversion

The ability of L6-CT neurons to elicit spontaneous nocifensive behavior (Fig. 1e) suggests that this pathway's activity is sufficient to induce a state of hypersensitivity and, contrary to a strict sensory discriminatory role, implicates S1 in regulating emotional-affective aspects of pain. We tested these hypotheses by assessing mechanical and noxious heat sensitivity and aversion as a function of L6-CT activation.

To quantify mechanical sensitivity, we used the von Frey test[43,44] to measure hindpaw withdrawal probabilities in response to mechanical stimulation over a range of stimulation forces (0.04–2 g), with and without optogenetic stimulation of L6-CT neurons in the contralateral S1HL (Fig. 2a, b). Using optogenetic laser intensities below the spontaneous paw lifting threshold (see Methods), L6-CT activation increased mechanical sensitivity over nearly the entire range of tested mechanical forces. Increased sensitivity to low-pressure punctate stimuli (0.04–0.4 g)[45] suggests an allodynic effect of L6-CT activity in S1 (i.e., a nociceptive response to typically innocuous stimuli), while increased sensitivity to nociceptor-activating mechanical forces (0.6–2 g)[46] represents a hyperalgesic effect. In addition to increasing mechanical sensitivity, L6-CT stimulation similarly amplified responses to noxious heat stimulation of the hindpaw (Supplementary Fig. 3). Mechanical and heat sensitivity remained unchanged in control animals (Ntsr1-Cre mice expressing EGFP in L6-CT; Supplementary Figs. 3, 4).

Does the L6-CT pathway also interact with a preexisting pain condition? To address this, we repeated von Frey experiments in a well-established inflammatory pain model induced by injection of Complete Freund's adjuvant (CFA)[47] in the left hindpaw, which induces allodynia and hyperalgesia (Supplementary Fig. 5). L6-CT activation in contralateral S1HL further increased mechanical sensitivity (Fig. 2c) in contrast to L6-EGFP controls in which sensitivity remained unchanged (Supplementary Fig. 4b, d). Taken together, L6-CT activation induced mechanical hyperalgesia and allodynia in naive animals, and furthermore exacerbated hyperalgesia and allodynia in an inflammatory model (Fig. 2b, c).

Because L6-CT activation can elicit a seemingly distressful and aversive experience (Fig. 1 and Supplementary Video 1), we sought to directly test the link between L6-CT activity in S1HL and negative affect by employing a real-time aversion paradigm[32]. Here, mice (L6-ChR2 and L6-EGFP controls) freely roamed a two-chamber setup during five baseline sessions to establish individual chamber preferences. In subsequent conditioning sessions, preferred chambers were paired with laser stimulation of the right S1HL cortex (schematic in Fig. 2e). Changes in the time spent in the preferred chamber were used to quantify the aversive effects of the stimulation. L6-ChR2 but not L6-EGFP control mice spent significantly less time in the laser-paired chamber relative to the time spent in the same chamber during the baseline session (i.e., without optogenetic stimulation) (Fig. 2f, g). The preference index for the paired chamber (PI) also dropped in L6-EGFP mice, indicating that the laser itself is optically aversive. However, the laser effect was significantly stronger in L6-ChR2 mice (Fig. 2h), who

developed an immediate and persistent avoidance of the stimulated chamber not observed in L6-EGFP controls (Supplementary Fig. 6). Together, these data show the potential of L6-CT-mediated changes in thalamocortical activity to cause hypersensitivity, amplified nociception, and aversion.

## Neuronal circuit mechanisms underlying the pronociceptive effect of L6-CT activation

We hypothesized that the pronociceptive effect of L6-CT stimulation during behavior arises from enhanced sensory signal flow through the thalamus and, in addition, might also involve cortico-cortical components, given that L6-CT neurons recruit local and translaminar cortical inhibitory neurons[8,9,42]. To assess the corticothalamic and cortico-cortical effects of L6-CT, we recorded thalamic and translaminar S1 single-unit spiking with silicon probes (Fig. 3a schematic and Supplementary Fig. 7) in combination with optogenetics and plantar

mechanical stimulation in anesthetized Ntsr1-Cre mice expressing ChR2 in L6-CT neurons in S1HL cortex.

We assessed single-unit spiking in three stimulation conditions (Fig. 3a): (1) Laser (L) stimulation of L6-CT neurons, (2) Mechanical (M) stimulation of the hindpaw, and (3) combined mechanical and laser (ML) stimulation. After pooling recordings across animals, we first screened for units with spiking responses significantly modulated relative to baseline in any of the three stimulus conditions (see Methods). The majority of units met this criterion (742/1018 in VPL and 751/842 in S1HL) and the remainder were not analyzed further. The response of each unit was further quantified as mean firing rate ($\bar{r}$) measured in a 1.5 s window after stimulus onset and modulation index (MI, −1 to 1) calculated by comparing stimulus-evoked spiking to baseline spiking. For VPL units, we also calculated burst probability (BP, 0–1), the ratio of burst events out of all observed responses, and response probability (RP, 0–1) per stimulus presentation.

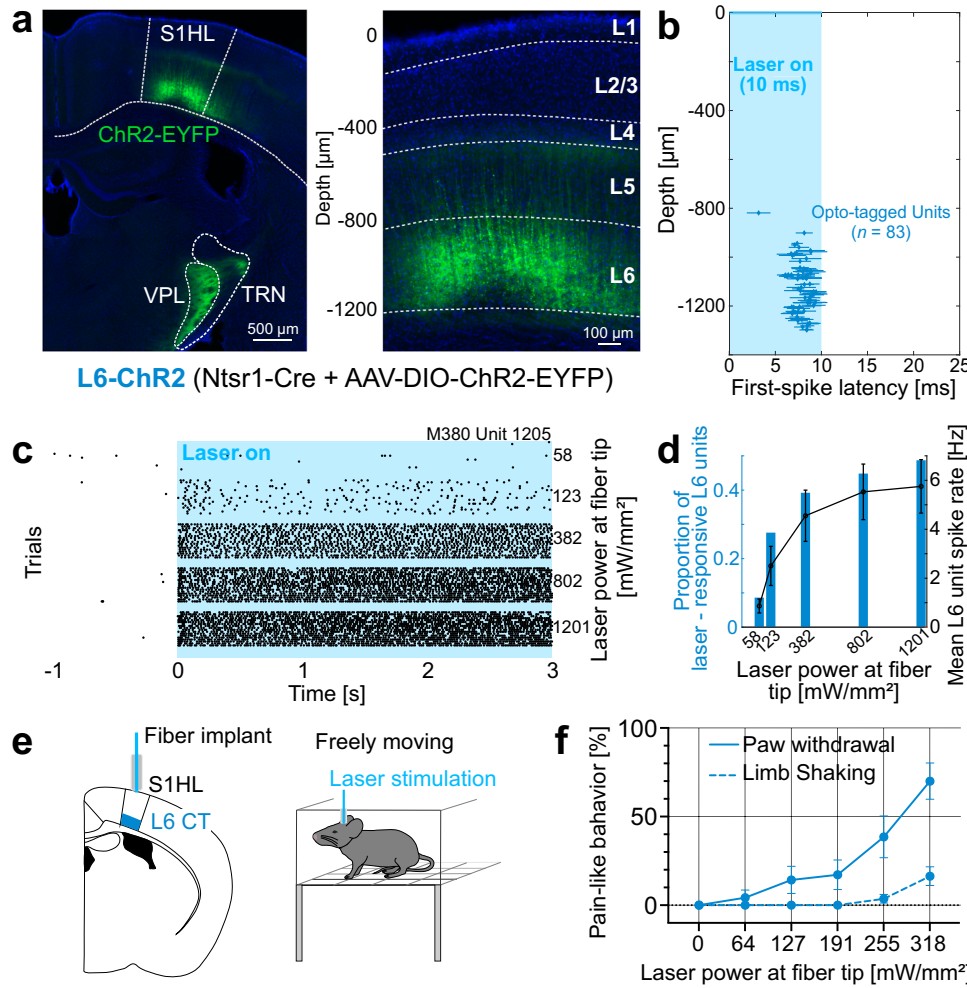

**Fig. 1 | Layer 6 corticothalamic (L6-CT) activation in the S1 hindlimb cortex (S1HL) elicits nocifensive behaviors in the absence of peripheral stimulation. a** ChR2-EYFP-expression (green) in S1HL of an L6-ChR2 mouse showing fluorescence in L6-CT neurons and their axons in the ventral posterolateral thalamus (VPL) and thalamic reticular nucleus (TRN). S1HL cortex (right panel) with L6-CT neurons depth-registered relative to S1HL layer borders (dashed lines, estimated based on soma sizes and densities using DAPI signals, blue). A representative example from n = 21 mice. **b–d** S1HL cortex silicon-probe recording from L6-ChR2 mice demonstrating optogenetic control of L6-CT units. **b** L6-CT units were identified based on their short-latency, low-jitter response to 10 ms light pulses (Supplementary Fig. 2). Each marker shows unit depth vs. mean ± SD latency to first evoked spike (pooled

from 3 animals, laser strength: 1201 mW/mm²). **c** Example rasters for a laser-responsive L6-CT unit (depth = 1205 μm). **d** Blue bars: fraction of laser-responsive L6 units (from n = 232 L6 units total) as a function of laser power. Black: corresponding mean spiking rate (mean ± SEM, n = 92 units; mean depth ± SD = 1194 ± 141 μm). Data from a representative experiment (n = 3 mice). **e** Schematic of fiber optic implant for optogenetic L6-CT stimulation in freely moving mice.
**f** Quantification of pain-like behaviors, paw lifting (blue solid line) and limb shaking (blue dashed line), elicited during 5 s of the optogenetic stimulation of S1HL L6-CT neurons in the absence of sensory stimulation (n = 11 mice, mean ± SEM). See also Supplementary Video 1. Source data for **b–d**, **f** are provided as a Source Data file.

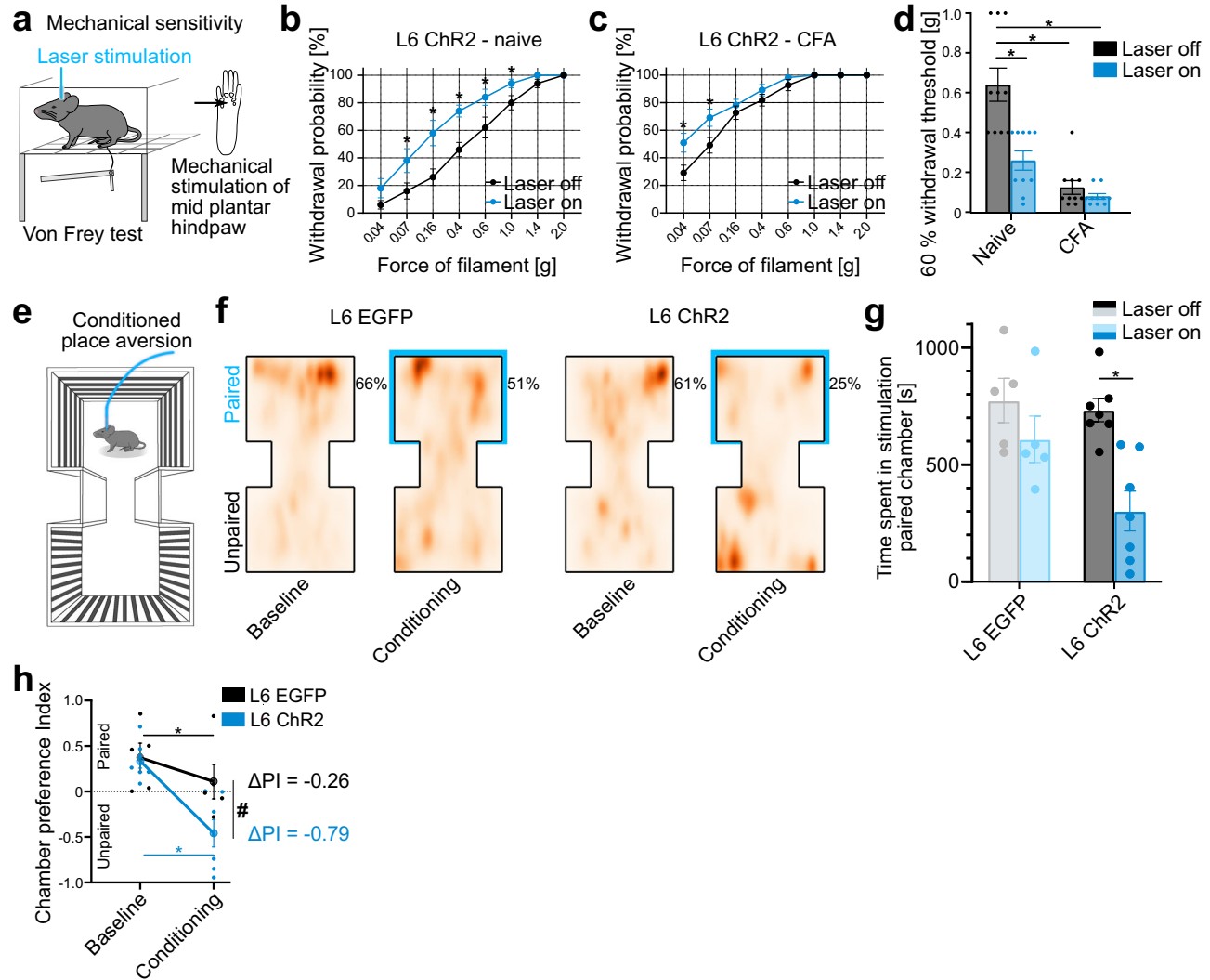

**Fig. 2 | Layer 6 corticothalamic (L6-CT) activation in the S1 hindlimb cortex (S1HL) increases mechanical sensitivity, exacerbates inflammatory allodynia, and induces aversion. a** Schematic of von Frey setup to quantify mechanical sensitivity in response to graded force stimulation of the hindpaw with and without optogenetic stimulation of L6-CT in contralateral S1HL. **b** Within-animal comparison of paw withdrawal probabilities in response to graded von Frey stimulation of the left hindpaw at baseline (black, Laser off) and during L6-CT laser stimulation (blue, Laser on, 5 s continuous pulse) stimulation in the contralateral S1HL of L6-ChR2 mice ($n = 10$, $p < 0.001$). L6-EGFP control animals in Supplementary Fig. 3b. **c** Same as in (**b**) but after animals were injected with Complete Freund's adjuvant (CFA) in the left hindpaw to induce paw inflammation ($n = 10$ mice). For comparison of pre- and post-CFA withdrawal probabilities, see Supplementary Fig. 5. **d** Comparison of filament forces that evoked paw withdrawal in 60% of the stimulation trials as a function of L6-CT activation without (Naive, $p < 0.0001$) and with (CFA) inflammation ($n = 10$ mice). Lower withdrawal thresholds indicate increased sensitivity. **e** Schematic of the experimental setup to measure real-time place aversion

as a function of optogenetic stimulation of L6-CT in the S1HL of naive animals. **f** Aggregated positional tracking heatmaps from L6-EGFP ($n = 5$) and L6-ChR2 ($n = 7$) mice across all respective baseline (no stimulation) and conditioning (20 Hz laser stimulation in S1HL cortex) sessions. Percentages show relative time spent in the laser-paired chamber (blue outlines). **g** Population analysis of total time spent in the laser-paired chamber at baseline (Laser off, black/gray) and during stimulation (Laser on, blue, 20 Hz laser stimulation in S1HL cortex) of L6-EGFP ($n = 5$) and L6-ChR2 ($n = 7$) mice. This avoidance behavior of L6-ChR2 animals persisted throughout the course of the experiment (Supplementary Fig. 6). **h** Average chamber preference indices (PI) for L6-EGFP ($n = 5$) and L6-ChR2 ($n = 7$) mice. A PI of 1 indicates a full preference, while a PI of −1 indicates full avoidance of the laser-paired chamber. PIs were significantly different between groups during laser stimulation, but not at baseline. * and # represent $p < 0.05$; 2**b**–**h**: Two-way repeated measures ANOVA with post hoc Bonferroni test. Exact $F$ and $p$ values are in Supplementary Table 1. Data were shown as mean ± SEM Source data for **b**–**d**, **f**–**h** are provided as a Source Data file.

## L6-CT activation enhances VPL output to the cortex

L6-CT stimulation enhanced VPL spiking in the absence of sensory stimulation (Fig. 3b–d, f, g), and both VPL $\bar{r}_L$ and $RP_L$ increased with stimulation strength (Supplementary Fig. 8a and Fig. 3g). L6-CT stimulation also decreased VPL bursting in favor of tonic spiking patterns (Supplementary Fig. 8b and Fig. 3h, i), in line with VPL depolarization during sustained CT activation as demonstrated in the whisker and visual thalamus[10,14,48]. These increases in both RP per trial and tonic spiking support a scenario in which L6-CT enhances VPL responses to somatosensory stimuli, as a mechanism for amplified nociception

observed in the behavior (Fig. 2). To test this hypothesis, we next assessed whether L6-CT activation at moderate laser intensities altered encoding of mechanical stimuli in a paired stimulation protocol.

Laser stimulation (L) of L6-CT and mechanical stimulation of the paw (M) alone modulated sizable fractions of VPL units (74 and 33%, L and M, respectively, Fig. 3b) with substantial overlap between the populations (69% of M-sensitive responded to L, 42% of L-sensitive responded to M, Fig. 3b), demonstrating that our probes targeted the paw-sensitive region of VPL and that S1 CT projections from ChR2-expressing L6 neurons overlapped with this region (probe location in

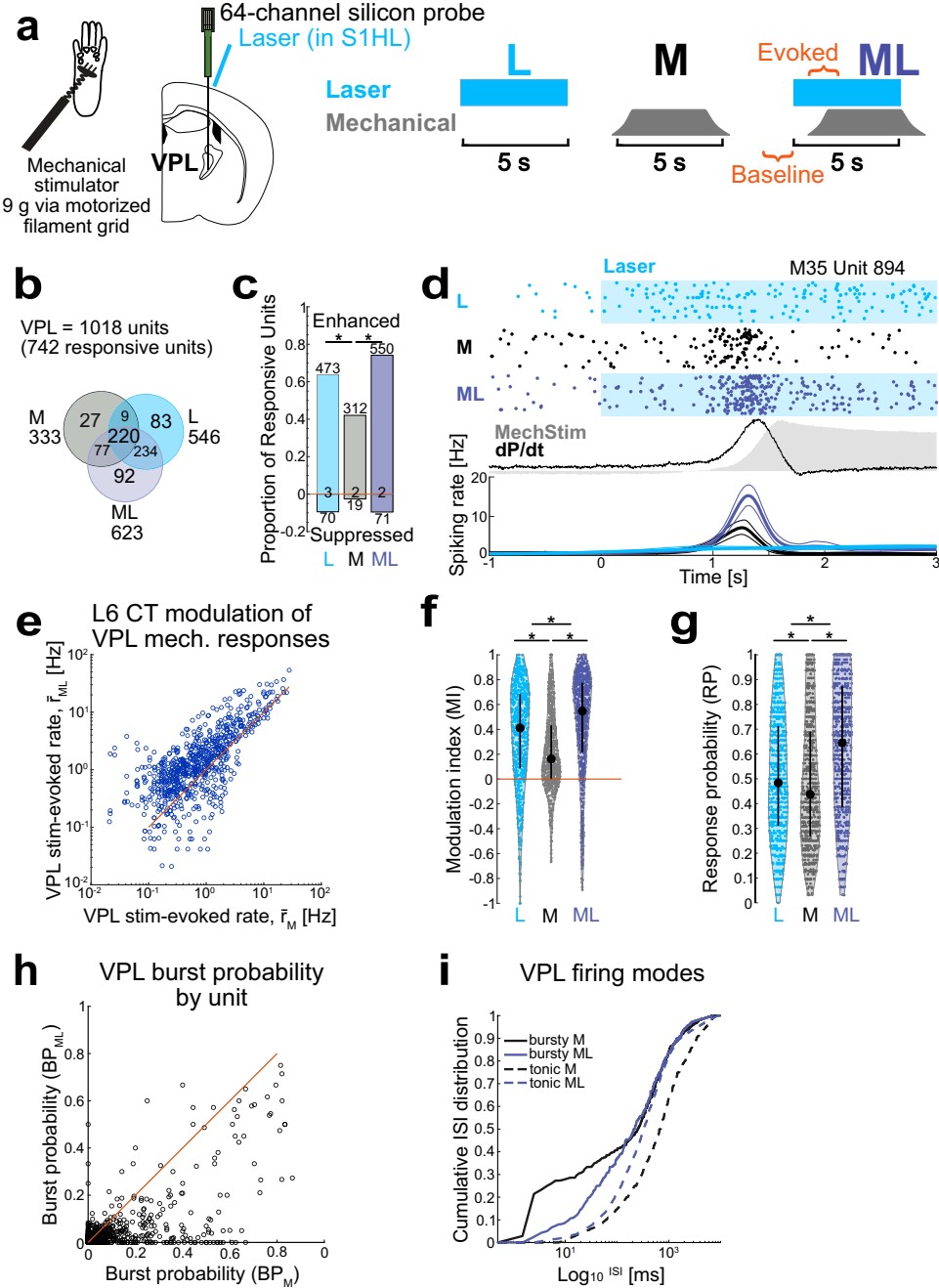

**Fig. 3 | Layer 6 corticothalamic (L6-CT) activation enhances ventral poster-olateral thalamus (VPL) spiking output. a** Left: Schematic of VPL silicon-probe recordings in anesthetized mice (*n* = 4). Right: stimulation protocol for three interleaved stimulation conditions: L6-CT activation (L), mechanical paw stimulation (M), and combined L6-CT + mechanical simulation (ML). Windows for unit spike analysis is indicated in orange. **b** Population overlap in L/M/ML conditions. 742/1018 VPL units were significantly modulated in at least one stimulus condition; 92 units were only responsive to ML. **c** Proportion and counts of enhanced and suppressed VPL units for each L/M/ML condition. For example, ~40% of the VPL units were enhanced by mechanical stimulation (M), while during the ML condition, 75% were enhanced and ~10% were suppressed. A minority of units had only changes in spike timing, not spike counts (counts at zero). **d** Example VPL single-unit responses to L, M, and ML stimulation. Top: raster examples; middle: mechanical pressure (gray shading) and first temporal derivative of pressure (dP/dt, black); bottom: corresponding PSTHs (bin size = 20 ms, smoothed with BARS method[76]). **e–g** ML condition enhances VPL responses. **e** Comparison of $\bar{r}_{ML}$ vs. $\bar{r}_M$ (0.54 **1.14** 2.81) Hz and (0.27 **0.67** 1.71) Hz, respectively, for all VPL units showing modulation in any condition (*n* = 742). The L6-CT-evoked difference in spiking rate ($\Delta\bar{r} = \bar{r}_{ML} - \bar{r}_M$): (0.02 **0.45** 1.19). Both **f** stimulus-evoked modulation index (MI) and

**g** response probability (RP) per trial varied by condition and were greatest in the ML condition. Black markers and bars show the median and IQR. **f** $MI_L$ (0.09 **0.41** 0.68), $MI_M$ (0.00 **0.16** 0.43), and $MI_{ML}$ (0.22 **0.55** 0.77) and **g** $RP_L$ (0.31 **0.48** 0.71), $RP_M$ (0.27 **0.44** 0.69), and $RP_{ML}$ (0.39 **0.65** 0.87). $MI_{ML} > MI_L > MI_M$ and $RP_{ML} > RP_L > RP_M$. **h** L6-CT activation decreases VPL burst probability (BP). For example, BP = 0.1 indicates 10% of spiking events were burst with 2 or more spikes. Scatter plot: BP of stimulus-evoked spiking, $BP_{ML}$ vs. $BP_M$, for each unit shown in **g** (638/742 units significantly different). $BP_{ML}$ (0 **0.016** 0.047) < $BP_M$ (0 **0.032** 0.144). **i** L6-CT activation regularizes stimulus-evoked spike timing in VPL. Cumulative interspike interval (ISI) distributions for M and ML conditions for bursty (BP > 0.1, solid line, *n* = 227) and tonic (BP ≤ 0.1, dashed, *n* = 515) units. Data were shown as the median per 1 ms bin for ISI distributions calculated separately for each unit. * represents *p* < 0.05; 3**c**: Two-sided $X^2$ test followed by Marascuillo procedure; 3**e**: two-sided Wilcoxon signed-rank or ZETA test; 3**f**, **g**: Friedmann test with post hoc Wilcoxon signed-rank test; 3**h**: two-sided McNemar's test (for change in BP) and two-sided Wilcoxon signed-rank test (for comparison of $BP_{ML}$ vs. $BP_M$). Exact *p* values are in Supplementary Table 1. Source data for **b–i** are provided as a Source Data file.

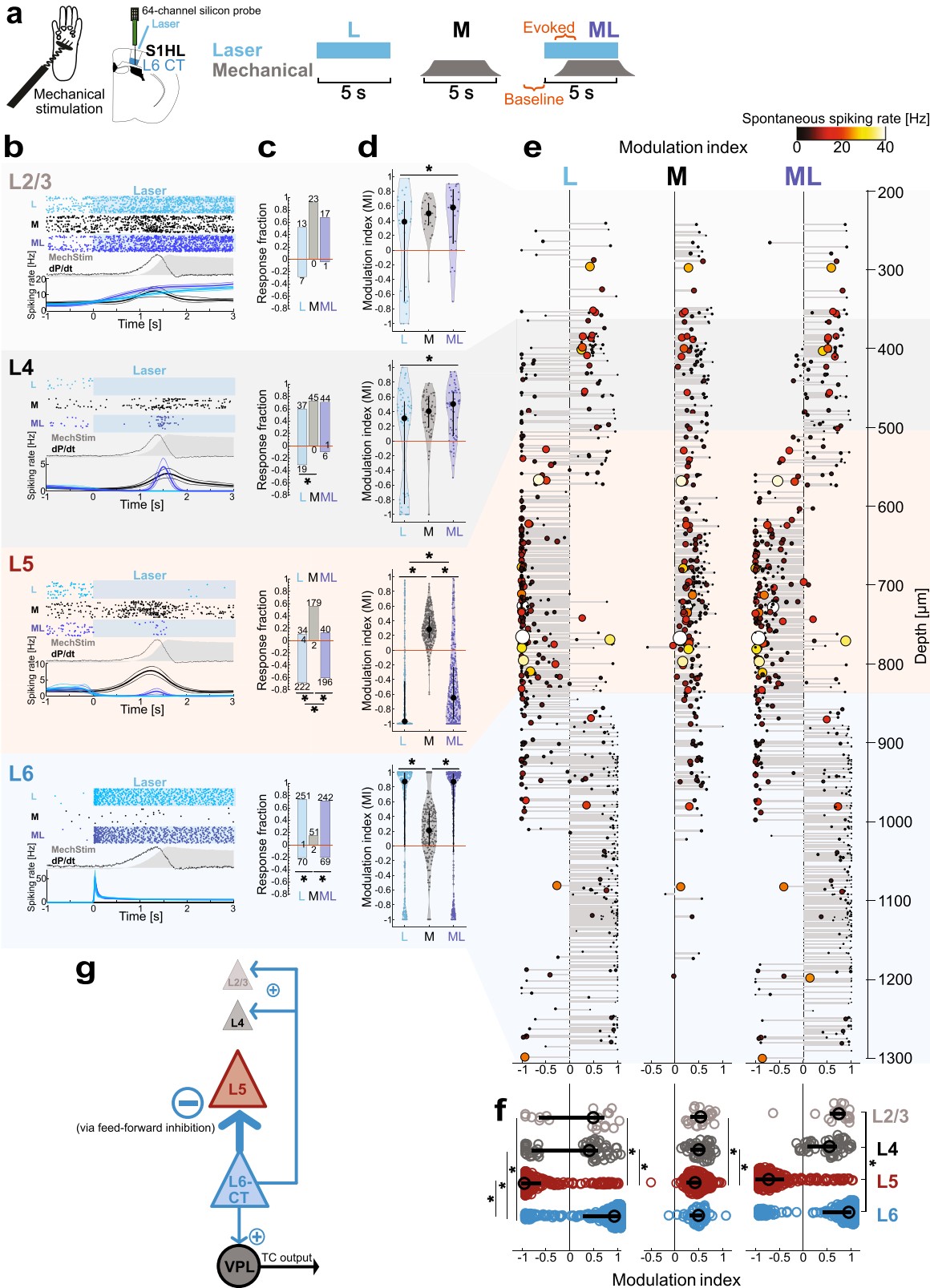

**Nature Communications** | (2023)14:2999

Supplementary Fig. 7). Paw stimulation alone evoked time-varying responses (Fig. 3d), with either phasic responses to initial changes in pressure (dP/dt) during stimulus onset, or less often, sustained responses throughout the duration of the stimulus. The direction of modulation by M relative to baseline was largely positive, in line with the established role of VPL as the main TC path for somatosensory information.

How does L6-CT activation modulate paw responses in VPL? Pairing L6-CT stimulation with paw stimulation (ML) enhanced the VPL representation of mechanical stimuli at the population and single-unit levels. In the ML condition, the fraction of responsive VPL units (61%) increased relative to both M and L conditions (33 and 54%, $p < 0.001$, McNemar's test)−including 12% of previously silent neurons which responded only to this combined condition. At the single-unit level,

**Fig. 4 | Layer 6 corticothalamic (L6-CT) activation suppresses spiking in layer 5 but enhances spiking in superficial layers. a** Upper: Experimental configuration for layer-resolved silicon-probe recordings in S1 hindlimb cortex (S1HL) of anesthetized mice ($n$ = 3). Lower: stimulation protocol for three interleaved stimulation conditions: L6-CT activation (L), mechanical paw stimulation (M), and combined L6-CT + mechanical simulation (ML). Time windows for unit spike analysis are indicated in orange. **b** Example responses for L, M, and ML conditions in L2/3 (light gray), L4 (gray), L5 (red), and L6 (blue). Top: raster examples; middle: mechanical pressure (gray shading) and first temporal derivative of pressure (dP/dt, black); bottom: corresponding PSTHs (bin size = 20 ms, smoothed with BARS method[76]). **c** Breakdown of layer-specific population responses to L/M/ML conditions showing proportion and counts of enhanced and suppressed units for each L/M/ML condition. A minority of units had only changes in spike timing, not spike counts (counts at zero). See Supplementary Fig. 8e for the population overlap of L/M/ML− sensitive units in each layer. **d** L6-CT stimulation enhances superficial layers' paw responses, but suppresses those of L5. Modulation indices pooled per condition and cortical layer. See Supplementary Fig. 8e for stimulus-evoked changes in spiking rates. Unit counts were $n$ = 25, 62, 323, and 341 for L2/3, L4, L5, and L6 units, respectively, pooled from three independent experiments. Data were shown as median and interquartile range. **e** Depth-resolved modulation indices (MI) for L2/3, L4, L5, and L6 units in S1HL for L, M, and ML conditions (left, middle, right rows, respectively; $n$ values as in **d**). Each unit is represented by one data point; the size and color of markers are proportional to the spontaneous spiking rate (black to yellow: low to high). Depth values are slightly jittered for visibility. Non-significantly modulated units are not shown. **f** $MI_L$, $MI_M$, and $MI_{ML}$ distributions with overlaid medians and first and third quartiles for significantly modulated units by layer (data replotted from **d** to facilitate comparison across layers). **g** Summary of functional effects of L6-CT activation on VPL and S1HL cortical layers; arrow widths proportional to experimental $MI_L$. * represents $p < 0.05$; **4c**: Two-sided $X^2$ test followed by Marascuillo procedure; **4d**, **f**: Friedman test with post hoc two-sided Wilcoxon signed-rank test; Exact $p$ values in Supplementary Table 1. Source data for **b**−**f** are provided as a Source Data file.

combined ML increased r̄, MI, and RP relative to both M and L control conditions (Fig. 3e and Supplementary Fig. 8). Estimating from the mean spiking rate and response probability observed across animals, CT activation approximately doubled mechanically-evoked thalamic output on the level of single VPL units (-2 AP/stim vs. 1.5 AP/stim vs 1 AP/stim, for ML, L, and M conditions, respectively). Aside from enhancing VPL response magnitude and probability, CT activation altered VPL spike timing statistics by shifting mechanically-evoked VPL spiking from burst to tonic mode (Fig. 3h, i). This effect depended on baseline firing mode and indeed served to normalize firing modes across units: (1) For tonic units (low BP), L6-CT activation shifted ISIs to lower values (Fig. 3i, dashed lines), reflecting increased firing rates. (2) In contrast, bursty (high BP) units were shifted to tonic mode, as CT activation decreased the fraction of short ISIs (Fig. 3i, solid lines). Thalamic tonic firing patterns likely depolarize cortical targets more effectively because they are less affected by the frequency-dependent depression at TC synapses[49].

In a separate set of experiments, we found that another thalamic target of S1 L6-CT, the posterior medial thalamic nucleus (POm), was more weakly and heterogeneously modulated by L6 activation, with lower median MI compared to VPL (Supplementary Fig. 9). Furthermore, the vast majority of POm units did not show significant mechanical responses, preventing further consideration of L6-CT modulation of POm mechanical responses under these experimental conditions.

Taken together, these findings suggest that the pronociceptive effect of L6-CT activation involves a large thalamic signaling component via VPL: under conditions of increased L6-CT activity, baseline feedforward thalamocortical signaling to the cortex is increased, suggesting a likely mechanism for the spontaneous pain-like behaviors, as well as enhanced thalamocortical representation of sensory stimuli during paw stimulation underlying hypersensitivity.

### Layer-specific modulation of S1 stimulus-evoked responses by L6-CT

As VPL is the primary thalamic route for somatosensory information to S1HL, we hypothesized that enhancement of thalamic responses by L6-CT could lead to concomitantly enhanced responses in the S1HL cortex. Additionally, L6-CT activity could alter S1HL responses more directly through cortical inhibitory networks. We therefore recorded depth-resolved spiking in S1HL layers 2–6 (Methods) during the paired stimulation protocol (Fig. 4a). The layer and cell-type-specific responses to mechanical, L6-CT, and paired stimulation for the entire pooled population of S1 neurons is summarized in Fig. 4b–e.

Consistent with direct optogenetic control of the L6 CT pathway shown in Fig. 1, the majority (74%) of L6 neurons (Fig. 4b–e, blue shading) were driven in the L condition, with spiking time-locked to the duration of the laser stimulus. In contrast, L6 units had weak

mechanical responses: only a small fraction (15%) of L6 units were significantly modulated in M, with modest $MI_M$ corresponding to low r̄$_M$ and low spontaneous spiking. This is consistent with weak and heterogeneous sensory modulation of L6 neurons reported in other sensory cortices[50,51].

We next examined the effect of L6-CT activation on other cortical layers. The degree of suppression and/or enhancement of neuronal responses by L6-CT was highly layer-specific, showing distinct effects on L5 versus L2/3 and L4 (Fig. 4b–f; see Supplementary Table 2 for comparisons across layers).

The most striking intracortical effect of L6-CT stimulation was a nearly complete abolishment of spiking in L5 units (Fig. 4b–e, red shading). While 56% of L5 units responded to paw stimulation, these responses largely disappeared with L6-CT stimulation. L6-CT stimulation also strongly suppressed spontaneous L5 spiking (69% negatively modulated, Fig. 4b, c). This suppressive effect of L6-CT on L5 has been described in the whisker cortex[52] and is mediated by feedforward inhibition[7–9,42]. We conclude that while L6-CT stimulation increases VPL output to the cortex (Fig. 3), this increased TC input is not sufficient to overcome the effect of intracortical feedforward inhibition on the L5 population.

In contrast, the effect of L6-CT activation on upper layers (L2/3 and L4) was weaker and more heterogeneous, for both spontaneous activity and modulation of mechanically-evoked responses (Fig. 4b–e, gray shadings). For both L2/3 and L4, L6-CT activation alone (L) enhanced and (suppressed) largely comparable fractions of units: L2/3: 52% (28%), L4: 60% (31%). Focusing on the subset of L2/3 and L4 neurons that were sensitive to paw stimulation (92 and 73%, respectively; Fig. 4c), both L2/3 and 4 units showed increased r̄$_{ML}$ relative to r̄$_M$ (Supplementary Fig. 8e), but the differences between M and ML conditions were not significant at the population level for either r̄ (Supplementary Fig. 8e) or MI (Fig. 4d). In sum, L6-CT activation weakly enhanced mechanical-evoked spiking of some units in superficial layers, consistent with increased VPL output (Fig. 3), and inhibited other units, consistent with L6-CT feedforward inhibition. This moderate net enhancement of superficial layers may contribute to L6-CT-mediated behavioral hypersensitivity (Fig. 2) during the processing of somatosensory stimuli.

Given the marked suppressive effect of L6-CT stimulation on L5 − particularly in contrast to the relatively mild enhancement of superficial layers−and the increasingly established role for L5 neurons in perception as well as pathologies such as chronic pain (reviewed in[53]), we reasoned that the L6−L5 functional interaction may contribute−in synergy with enhanced VPL-S1 TC signaling−to the modulation of pain-like behaviors. To test this hypothesis, we employed cell-type-specific optogenetic control of L5 neurons in behaving mice and studied the function of L5 activation and suppression on sensitivity, analgesia, and aversion.

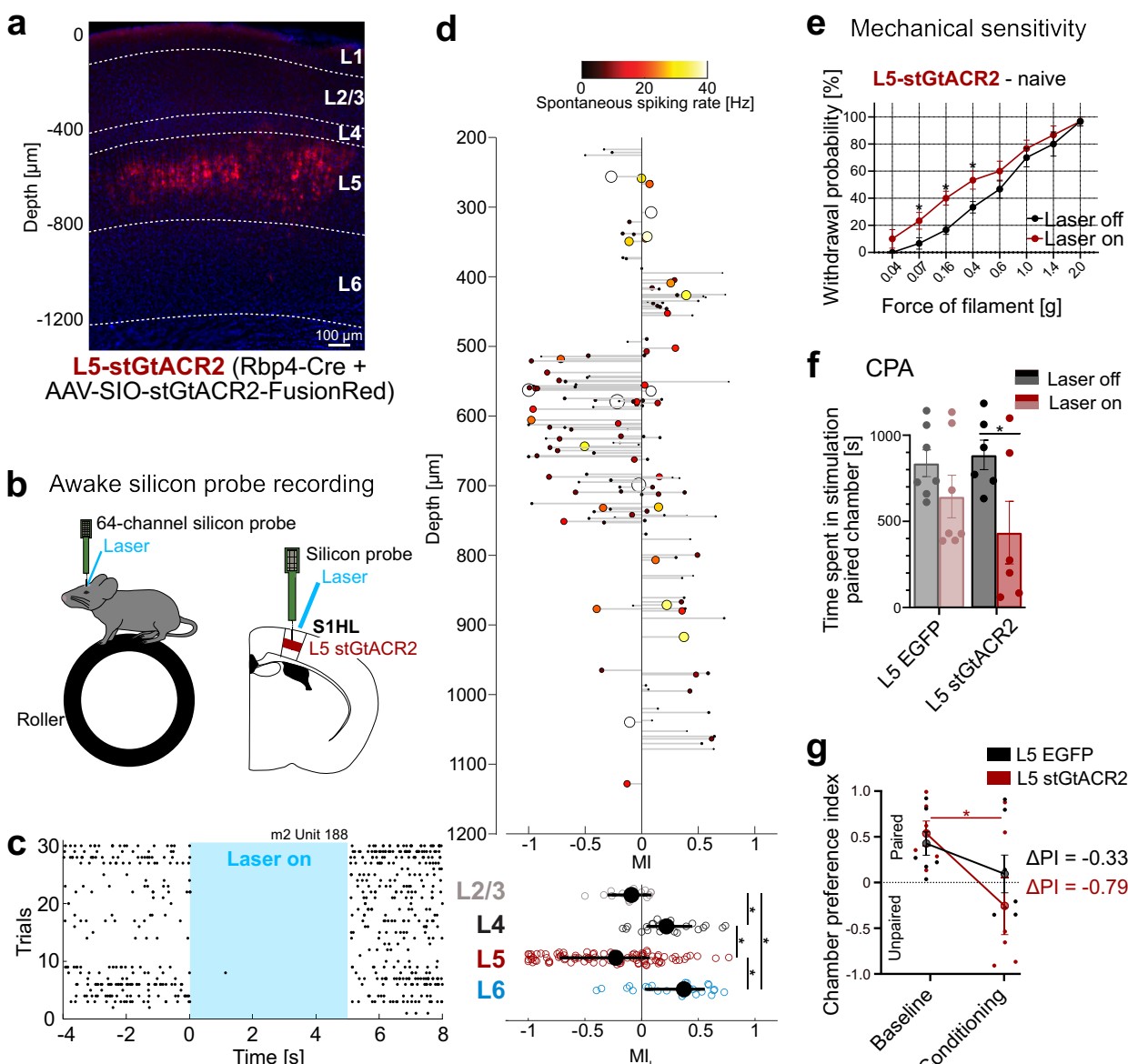

**Fig. 5 | Layer 5 (L5) inhibition in the S1 hindlimb cortex (S1HL) reproduces pronociceptive effects of Layer 6 corticothalamic (L6-CT) activation.**
**a** stGtACR2-FusionRed-expression (red) in L5 S1HL of an Rbp4-Cre mouse showing fluorescence in L5 neurons, S1HL layer borders (dashed lines), estimated based on soma sizes and densities using DAPI signals (blue). A representative example from $n = 15$ mice. **b** Schematic of silicon-probe recordings in S1HL in a head-fixed awake mouse (upper); example raster demonstrating inhibition of spiking in an example L5 unit (lower). **c** Example raster for a laser-suppressed L5 stGtACR2 unit (depth = 589 μm). **d** Upper panel: depth-resolved light modulation indices ($MI_L$) for L2/3, L4, L5, and L6 units in S1HL during optogenetic inhibition of L5 ($n = 3$ pooled experiments, 285 units). Each unit is represented by one data point; the size and color of markers are proportional to the spontaneous spiking rate (black to yellow: low to high). Depth values are slightly jittered for visibility. Non-significantly modulated units are not shown (122/285). Lower panel: $MI_L$ distributions for significantly modulated units by layer. **e** Within-animal comparison of paw withdrawal probabilities in response to graded von Frey stimulation of the hindpaw at baseline (black, laser off) and during optogenetic inhibition (red, laser on, 5 s continuous pulse) in the contralateral S1HL of L5-stGtACR2 mice ($n = 6$). **f** Conditioned place aversion (CPA) test. Population analysis of total time spent in the laser-paired chamber at baseline (Laser off, black/gray) and during inhibition (Laser on, red 20 Hz laser stimulation in S1HL cortex) of L5-EGFP ($n = 7$) and L5-stGtACR2 ($n = 6$, $p = 0.006$) naive mice. **g** Average chamber preference indices (PI) for L5-stGtACR2 ($n = 6$) and L5-EGFP ($n = 7$) mice. A PI of 1 indicates a full preference, while a PI of −1 indicates full avoidance of the laser-paired chamber. PIs were not significantly different between groups during laser stimulation or at baseline. * represent $p < 0.05$; 5d: two-sided Wilcoxon signed-rank test; see Supplementary Table 1 for population medians and first and third quartiles per layer and condition.; 5e–g: Two-way repeated measures ANOVA with post hoc Bonferroni test. Exact $F$ and $p$ values are in Supplementary Table 1. Data were shown as mean ± SEM. Source data for **c**–**g** are provided as a Source Data file.

## L5 inhibition in S1HL reproduces the pronociceptive phenotype of L6-CT stimulation, while L5 stimulation is antinociceptive

Because L6-CT stimulation induced hypersensitivity and, at the same time, strongly suppressed L5 neurons, we asked whether the L6-CT behavioral modulation could be in part due to the suppression of L5. We, therefore, directly suppressed L5 using the inhibitory opsin stGtACR2[54] (characterization in Supplementary Fig. 10) in the Rpb4-

Cre line (Fig. 5). We found that suppression of L5 leads to a net enhancement of L6 in S1HL in awake animals (10% suppressed vs. 42% activated, Fig. 5d), in close agreement with recent findings from the primary auditory cortex[55]. On the behavioral level, direct optogenetic suppression of L5 neurons had similar effects as L6-CT stimulation, namely increased mechanical sensitivity (Fig. 5e) and aversion in the CPA paradigm (Fig. 5f, g, heat sensitivity and CPP in Supplementary

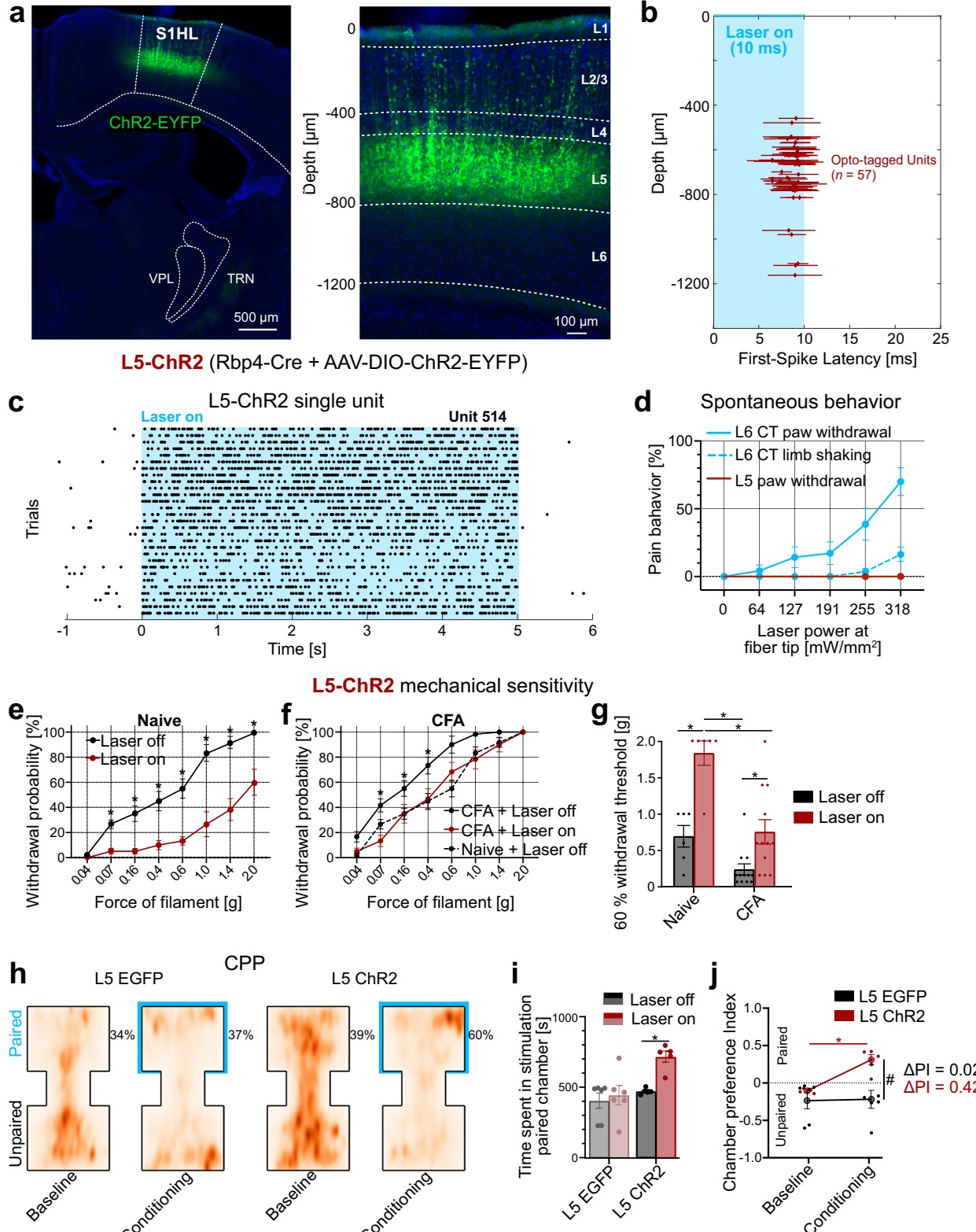

**L5-ChR2** (Rbp4-Cre + AAV-DIO-ChR2-EYFP)

**L5-ChR2** mechanical sensitivity

Fig. 11). Thus, L5 inhibition reproduces the hypersensitivity and aversion phenotype we observed for L6-CT stimulation.

Based on the pronociceptive effect of L5 inhibition, we next reasoned that stimulation of L5 in S1HL might have an opposite, antinociceptive effect. To directly test this possibility, we next optogenetically stimulated L5 neurons in S1HL in channelrhodopsin-2-EYFP ("L5-ChR2") or EGFP ("L5-EGFP") expressing Rbp4-Cre mice (Fig. 6a; for

EGFP-controls, see Supplementary Fig. 1a). Silicon-probe recordings in anesthetized L5-ChR2 mice validated optogenetic drive of short-latency and low-jitter spiking L5 activity (Fig. 6b) that persisted throughout the 5 s laser stimulus (Fig. 6c).

As with the L6-CT pathway, we first assessed whether L5 activation in the S1HL cortex affected spontaneous behavior in fiber-implanted, freely moving animals. At all tested laser intensities (0−318 mW/mm²),

**Fig. 6 | Layer 5 (L5) activation in the S1 hindlimb cortex (S1HL) is anti-nociceptive. a** ChR2-EYFP-expression (green) in S1HL of a L5-ChR2 mouse showing fluorescence in L5 neurons. S1HL cortex (right panel) with L5 neurons depth-registered relative to S1HL layer borders (dashed lines, estimated based on soma sizes and densities using DAPI signals, blue). A representative example from $n = 19$ mice. **b**, **c** Representative S1HL cortex silicon-probe recording from an L5-ChR2 mouse demonstrating optogenetic control of the L5 pathway. **b** L5 units were identified based on their short-latency, low-jitter response to 10 ms light pulses (1201 mW/mm²). Each marker shows unit depth vs. mean ± SD latency to the first evoked spike ($n = 32$ out of 203 units recorded at a depth of L5, data from $n = 2$ mice). **c** Example raster plot of one unit from b in response to 5 s laser stimulation (1201 mW/mm²). **d** Optogenetically-evoked pain-like behaviors (i.e., paw shaking and withdrawal) were absent in the case of L5 stimulation (red, $n = 6$). L6-CT (blue, $n = 11$, replotted from Fig. 1f). Behavioral responses were considered in the 5 s optogenetic stimulation period, and paw lifting (solid line), and limb shaking (dashed line). See also Supplementary video 2. **e** Within-animal comparison of paw with-drawal probabilities in response to graded von Frey stimulation of the left hindpaw at baseline (black, laser off) and during laser stimulation (red, laser on, 5 s continuous pulse) in the contralateral S1HL of L5-ChR2 mice ($n = 6$, $p < 0.001$). L5-EGFP control animals in Supplementary Fig. 4a. **f** Same as in (**e**) but after animals were injected with Complete Freund's adjuvant (CFA) in the left hindpaw to induce paw inflammation ($n = 12$). Comparison between pre- and post-CFA in Supplementary Fig. 5. **g** Comparison of filament forces that evoked paw withdrawal in 60% of the stimulation trials as a function of L5 activation without (Naive; $n = 6$ mice, $p = 0.0004$) and with inflammation (CFA; $n = 12$ mice, $p = 0.03$). Higher withdrawal thresholds indicate decreased sensitivity. **h** Conditioned place preference (CPP) test. Aggregated positional tracking heatmaps from L5-EGFP ($n = 6$) and L5-ChR2 ($n = 5$) mice across all respective baseline (no stimulation) and conditioning (20 Hz laser stimulation of S1HL cortex) sessions. Percentages show relative time spent in the laser-paired chamber (blue outlines). Animals were injected with CFA (see Methods) one day before the first baseline session. **i** Population analysis of total time spent in the laser-paired chamber at baseline (Laser off, black/gray) and during stimulation (Laser on, red 20 Hz laser stimulation in S1HL cortex) of L5-EGFP ($n = 6$) and L5-ChR2 ($n = 5$) mice. **j** Average chamber preference indices (PI) for L5-ChR2 ($n = 5$) and L5-EGFP ($n = 6$) mice. A PI of 1 indicates a full preference, while a PI of −1 indicates full avoidance of the laser-paired chamber. PIs were significantly different between groups during laser stimulation, but not at baseline. * and # represent $p < 0.05$; 6**e**–**j**: Two-way repeated measures ANOVA with post hoc Bonferroni test. Exact $F$ and $p$ values are in Supplementary Table 1. Data were shown as mean ± SEM. Source data for **b**–**j** are provided as a Source Data file.

none of the L5-ChR2 animals showed optogenetically-elicited noci-fensive behavior, in stark contrast to the observations in the L6-ChR2 animals (Fig. 6d and Supplementary Video 2). L5 activation in S1HL also did not induce an aversive phenotype (Supplementary Fig. 12). How does optogenetic control of L5 modulate paw sensitivity to acute sti-mulation? Quantification of mechanical sensitivity using the von Frey test revealed that L5 activation significantly decreased mechanical sensitivity over nearly the entire range of tested mechanical forces (Fig. 6e), while L5 suppression increased mechanical sensitivity (Fig. 5). Sensitivity remained unchanged in L5-EGFP control animals (Supple-mentary Fig. 4a).

These effects in naive mice suggest an anti-nociceptive function of the L5 pathway. To test this directly, we repeated the mechanical sensitivity measurements in the CFA inflammatory pain model. In CFA-injected L5-ChR2 animals, laser stimulation of the S1HL decreased mechanical sensitivity (Fig. 6f), an effect not present in L5-EGFP control animals (Supplementary Fig. 4c). Furthermore, we observed that L5 stimulation in CFA mice completely reverses mechanical sensitivity back to pre-CFA sensitivity (Fig. 6f, g), suggesting that L5 stimulation can ameliorate inflammatory pain. To further explore this possibility, we conducted real-time place preference tests, in which mice freely explored two chambers, of which the non-preferred chamber was longitudinally paired with optogenetic stimulation of S1HL. L5-ChR2 mice, but not L5-EGFP controls, showed real-time conditioned place preference (CPP) for the stimulation-paired chamber (Fig. 6h–j). Thus, L5 stimulation reduces sensitivity in naive animals and causes a place preference in the inflammation model, while suppression of L5 causes hypersensitivity and aversion.

### S1 bidirectionally modulates paw sensitivity via L6-CT and L5
We next directly compared the effects of L5 and L6-CT stimulation across animals and stimulation forces, illustrating that L5 reduces sensitivity and L6-CT increases sensitivity for a given mechanical sti-mulation force (Fig. 7a).

As a single readout for overall mechanical sensitivity, we com-puted areas under the curves (AUC) from all mechanical sensitivity measurements, which allowed us to directly compare the modulatory functions of cortical pathway-specific optogenetic stimulation on sensitivity in the naive and CFA-inflamed mouse models. Compared to baseline levels (Laser off), high AUC values correspond to hypersen-sitivity and were caused by L6-CT stimulation and by paw inflammation in CFA animals. In contrast, low AUC values indicate reduced sensitivity and were caused by L5 stimulation (Fig. 7b), demonstrating that L6-CT and L5 in S1 can modulate paw sensitivity in opposite directions (Fig. 7c). In summary, this bidirectional modulation of mechanical

sensitivity by S1HL output pathways complements the affective behavioral results obtained in this study (see overview in Supple-mentary Table 3), together supporting pronociceptive effects follow-ing L6-CT activation and antinociceptive effects following L5 activation.

## Discussion
Pain is an inherently context-dependent phenomenon. Competing needs, psychosocial factors, and cross-modal sensory interactions can exacerbate or ameliorate the subjective experience of pain[16,56,57], and such contextual effects point to central neural mechanisms for "top-down" control over pain perception. Here, we investigated how two distinct top-down pathways from the S1HL cortex modulate somato-sensation in behaving mice (Fig. 8), revealing that L6-CT activation not only increases somatosensory sensitivity but also evokes aversion and nocifensive behavior even in the absence of peripheral stimulation, whereas L5 activation has an opposing, antinociceptive effect. This bidirectional control of pain-related behaviors was effective in both naive conditions and inflammatory pain models.

The marked aversive effects of L6-CT activation were unexpected, as existing research has primarily studied L6-CT in purely sensory processing contexts and left unresolved how L6-CT functions link to subjective sensory experiences and to behavior. It is broadly estab-lished that L6-CT pathways endow sensory systems with "beneficial" information processing functions, such as controlling sensory gain and tuning selectivity[7,8,11,52,58] and mediating burst-tonic transitions in the thalamus[10,12,59]. Our observation that S1HL L6-CT activation can cause aversive hypersensitivity reveals a nociceptive potential for S1 L6-CT to gain control. Therefore, these results expand the function of L6-CT, suggesting that, contingent upon L6-CT output level, gain control follows a functional continuum spanning both increased sensitivity and aversive hypersensitivity—even exacerbating preexisting hyper-sensitivity in inflammatory pain and triggering spontaneous pain-like behavior.

What are the possible circuit mechanisms underlying L6-CT-mediated hypersensitivity? To assess corticothalamic[1,10,14] and cortico-cortical effects[8,9,42] of L6-CT activation, we conducted silicon-probe recordings in VPL and S1 to measure how L6-CT activity modulates thalamocortical representations of mechanical stimuli. We found that L6-CT enhances VPL spiking and modulates S1 in a layer-specific manner, with the most drastic effect being the suppression of L5. This dual effect of L6-CT to suppress L5 while enhancing the thalamus has been described in the whisker TC system, but not in the context of behavior or pain[52]. Notably, our VPL recordings revealed several par-allel mechanisms favoring increased thalamocortical sensory

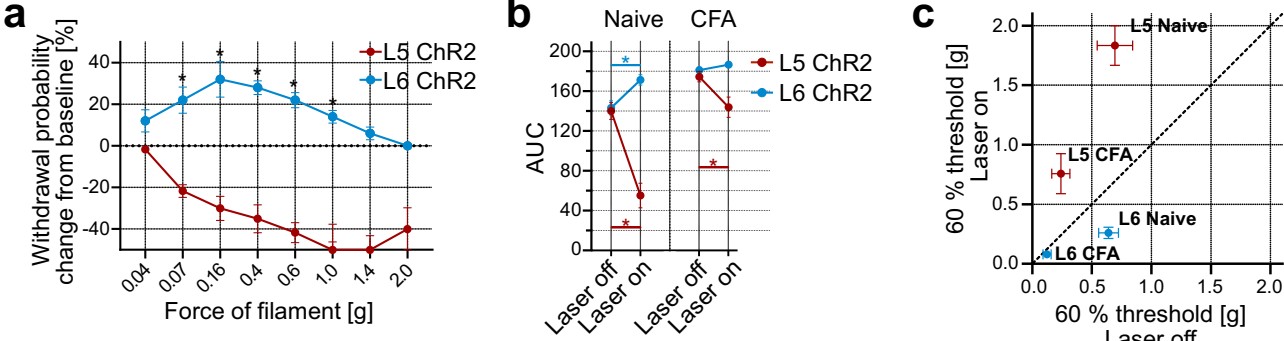

**Fig. 7 | Bidirectional modulation of sensory gain and nociception via S1 Layer 6 corticothalamic (L6-CT) and Layer 5 (L5). a** Withdrawal probabilities during L5 (red, *n* = 6 mice) or L6-CT (blue, *n* = 10 mice) optogenetic stimulation (5 s continuous laser above S1 hindlimb cortex (S1HL)) as a function of filament force, expressed as changes from baseline measurements (stippled line, Laser off trials). *p* < 0.001. **b** Comparison of mechanical sensitivity, computed from areas under the curves (AUC) from naive and Complete Freund's adjuvant (CFA)-inflamed mice, with (Laser on) and without (Laser off) activation of L5 and L6-CT neurons in S1HL. Increased AUCs correspond to increased sensitivity (hypersensitivity) and decreased AUCs correspond to decreased sensitivity (hyposensitivity). Numbers of mice: Naive L5 *n* = 6, *p* < 0.001; Naive L6-CT *n* = 10, *p* < 0.001; CFA L5 *n* = 12,

*p* < 0.001; CFA L6-CT *n* = 10, *p* = 0.84. **c** Comparison of filament forces that evoked paw withdrawal in 60% of the stimulation trials measured in naive and CFA-inflamed mice with (Laser on) and without (Laser off) activation of L5 and L6-CT neurons in S1HL. Data points below the diagonal (dashed line) correspond to increased sensitivity (hypersensitivity) and above correspond to decreased sensitivity (hyposensitivity). Numbers of mice: Naive L5 *n* = 6; L6-CT *n* = 10; CFA L5 *n* = 12; L6-CT *n* = 10. * represents *p* < 0.05; 7a, b: Two-way repeated measures ANOVA with post hoc Bonferroni test. Exact *F* and *p* values for all statistical tests are given in Supplementary Table 1. Data were shown as mean ± SEM. Source data for **a**–**c** are provided as a Source Data file.

transmission. L6-CT activation not only enlarged the fraction of sensory-encoding units in VPL, but also increased spike output on the level of individual units, and in parallel, both favored tonic spiking mode and inhibited non-responding units. Another thalamic target of L6-CT is POm[20] which has been implicated in pain processing[60]. L6-CT stimulation also modulated POm activity, albeit more weakly and heterogeneously compared to VPL. Taken together, the thalamic effects of L6-CT activation predict an enhanced signal-to-noise ratio of sensory output to the cortex. Consistent with an increase in thalamic signaling, at the cortical level, L4 and L2/3 responses to mechanical stimuli increased. L6-CT recurrent excitation via the thalamus likely contributes to the net enhancement of L2/3 and L4[61] and we cannot fully resolve the recurrent component in L2/3 and L4 in the present study. Nevertheless, this enhancement of the VPL – L4 – L2/3 axis suggests that L6-CT can amplify sensory transmission through the canonical circuitry of the lemniscal pathway of the lateral pain system, which carries body signals to S1[34]. Furthermore, at the same time as sensory transmission to S1 is amplified, antinociceptive L5 output from the cortex is suppressed.

The near complete suppression of L5 neurons' spontaneous and sensory-evoked spiking likely comes about from L6-CT recruitment of inhibitory circuits, as demonstrated in the barrel and visual cortices. Based on this work, excitatory effects of L6-CT neurons are via direct glutamatergic synapses, while the suppression of L5 is mediated by feedforward inhibition via L6-CT's selective recruitment of deep-layer, fast-spiking parvalbumin-positive inhibitory neurons[7–9,42,52].

We hypothesized that the L6-CT-mediated suppression of L5 contributes to increased nociception. Consistent with this hypothesis, we found that direct optogenetic suppression of L5 increased sensitivity and caused place aversion. Conversely, L5 stimulation reduced mechanical sensitivity, reversed inflammatory allodynia, and caused place preference in animals with pharmacologically-induced inflammatory allodynia (for a summary of the outcomes of the main behavioral manipulations, see Supplementary Table 3). Together these findings suggest that the pronociceptive function of L6-CT results from combined suppression of L5 and enhancement of thalamocortical excitability via enhancing the VPL – L4 – L2/3 axis (Fig. 8).

These results potentially have clinical value, such that positive effects on pain states may be achieved by preventing L6-CT hyperactivity and/or by enhancement of L5 activity in the S1 cortex. Based on the corticothalamic anatomy of the S1 limb area, this antinociceptive action of L5 is unlikely to directly involve VPL, which does not receive L5 input[20], but may rather be mediated by descending modulation of subcortical L5 target circuits (Fig. 8)—e.g., in POm, zona incerta, periaqueductal gray, or spinal cord—which have been associated with pain[60,62,63]. Untangling the downstream effects of L5 stimulation may generate highly specific interventional strategies for pain management.

Behavioral determinants and circuit mechanisms controlling L6-CT output pathways are only beginning to be studied, e.g., refs. [64,65], let alone those that potentially lead to L6 hyperactivity or synchronization. Interestingly, there appears to be a strong link between pain states and increased S1 activity[66,67] and cortical synchronization[31,32,68]. Optogenetic stimulation tends to strongly activate and synchronize neuronal activity[69], an effect which likely helped to uncover—but also likely exaggerated—adverse effects of L6-CT gain control and antinociceptive effects of L5 as described in this study. Nonetheless, these results demonstrate the potential of cortical output pathways to turn sensory signals into painful experiences or dampen nociception, even though it is currently not known what drives L6-CT neurons under physiological or pathological conditions[61].

While our study did not directly focus on the interactions between L6 and L5, our data concurs with recent research in the auditory cortex that showed that optogenetic L5 stimulation inhibited L6[55]. In line with this L5 feedforward suppression of L6, we find that L5 inhibition increases L6 spiking. In turn, L6 stimulation inhibits L5, as shown here and in previous work[7,52]. Generalizing these results suggests that these two distinct cortical output layers may 'compete' with one another, and that the result of this competition influences sensory perception, including pain sensitivity and affect. Changes to either L5 or L6-CT would not only alter perception directly—with respect to a single cortical output pathway—but would also tip the balance between cortical output pathways through mutual inhibition. For example, in the context of the present study, L6-CT is pronociceptive directly through increased sensory gain from the periphery, and indirectly via its inhibition of the antinociceptive L5 pathway.

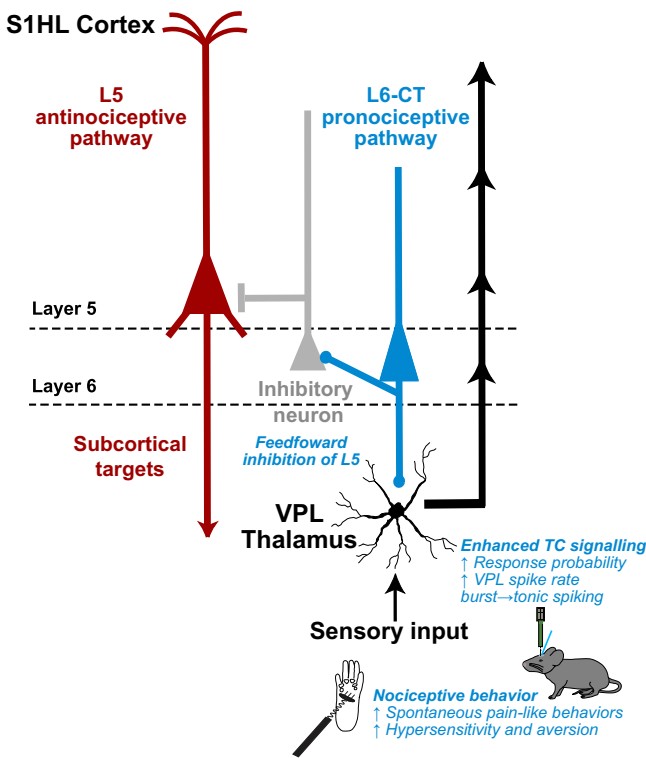

**Fig. 8 | Summary of bidirectional modulation of sensory gain and nociceptive behavior by S1 hindlimb cortex (S1HL) layer 5 and Layer 6 corticothalamic (L6-CT) neurons.** Main findings of this study in the context of known thalamocortical circuitry of the S1 cortex[6,9,20]. L6-CT activity increases sensory gain and induces nociception via enhancement of thalamocortical transmission through L6-CT glutamatergic projections to ventral posterolateral thalamus (VPL) and feedforward inhibition of L5 neurons.

In conclusion, this study reveals dual modulation of sensitivity and nociception via two specific populations of cortical output neurons in S1, underscoring the potential of cortical output pathways as therapeutic targets. The role of S1 in pain has been debated[35], as targeting S1 has been shown to both beneficially alter pain trajectories but also enhance nociceptive responses[29,30,70,71]. Moreover, previous causal studies affected bulk S1 activity across all layers and cell types and were thereby unable to functionally disambiguate the opposing roles of the two cortical output pathways, as revealed in this study. Understanding how the balance of corticofugal output is maintained in health and altered in disease is a promising direction for future studies.

## Methods

### Ethics statement
All experimental procedures were approved by the local governing body (Regierungspräsidium Karlsruhe, Germany, approval numbers: 35-9185.81/G-29/16, 35-9185.81/G-70/21, T-39-20, and 35-9185.82/A-8/20) and performed according to their ethical guidelines.

### Animals
Mice (male and female, 7–16 weeks of age) were housed with food and water ad libitum on a 12 h light/dark cycle (housing conditions 20–22 °C, 40–65% humidity).

### Mouse lines
**Layer 6 optogenetic stimulation.** "Ntsr1-Cre" (B6.FVB(Cg)-Tg/(Ntsr1-cre)GN220Gsat/Mmucd)

"Ntsr1-Cre-ChR2-EYFP"; crossbreed between "Ntsr1-cre" x "Ai32" (B6.FVB(Cg)-Tg/(Ntsr1-cre)GN220Sat/Mmucd x B6.129S-Gt(ROSA)26Sortm32(CAG-COP4*H134R/EYFP).

**Layer 5 optogenetic stimulation.** "Rbp4-Cre" (B6.FVB/CD1-Tg(Rbp4-cre)KL100Gsat/Mmucd)

### Virus injection and optical fiber implantation
Ntsr1-Cre and Rbp4-Cre mice were stereotaxically injected with either an excitatory opsin (AAV2-EF1a DIO-ChR2(H134R)-EYFP-WPRE-pA, $5.7 \times 10^{12}$ vg/ml, Zürich vector core), an inhibitory opsin (AAV1-hSyn-SIO-stGtACR2-FusionRed, $5 \times 10^{11}$ vg/ml, Addgene) or a control virus (AAV2-hSyn-DIO-EGFP, 100 μL at titer $\geq 3 \times 10^{12}$ vg/mL, Addgene). A subset of the optogenetic L6-CT experiments was done in Ntsr1-Cre-ChR2-EYFP mice, which were not virus injected. We observed no differences in any of the measurements and pooled Ntsr1-Cre-ChR2-EYFP mice and virus-injected Ntsr1-Cre experiments. Virus expression time was between 3-4 weeks.

For the injection and implantation, mice were placed in a stereotaxic frame (Kopf Instruments) and anesthetized with 1.2–2.0 vol% isoflurane in medical oxygen at a flow rate of 0.8 L/min, while keeping body temperature at 39 °C. Carprofen (CP-Pharma) was administered subcutaneously (5 mg/kg) and Lidocaine (Xylocaine 1%, Aspen Pharma) was injected under the scalp and around fixation ear bars for local anesthesia. Eyes were covered with Bepanthen ointment (Bayer) to prevent eye drying during the surgery. After ensuring the absence of tail and toe pinch reflexes, the skin was opened with a midline incision, the periosteum and aponeurotic galea was removed to visualize anatomical reference points (bregma and lambda), and the head was aligned to the stereotaxic frame. Small craniotomies were drilled above the S1HL area and viral particle solutions were injected into two sites within S1HL with calibrated glass micropipettes (Blaubrand; IntraMARK) at the following coordinates relative to bregma (AP, ML) and pia mater (DV):

First injection: ML = +1.4 mm, AP = −0.46 mm; sec injection: ML = +1.5 mm, AP = −0.94 mm. Ntsr1-Cre mice were injected at a depth of −0.9 and −1.0 mm, while Rbp4-Cre mice were injected at a depth of −0.7 and −0.8 mm (100 nl at each depth, followed by a waiting period of 10 min before relocating the injecting pipette).

Chronic optical fiber implants (200 μm diameter, numerical aperture of 0.39, Thorlabs GmbH) were placed on the dura above the S1HL (ML = +1.5 mm, AP = −0.94 mm), and the ceramic ferrule encapsulating the optical fiber was fixed to the skull with dental cement. To minimize laser light leakage, the cement was colorized in black and the mating sleeve used during the experiments was covered with black tape. Mice were kept between three and four weeks for optimal viral expression before experiments.

### Histology and immunohistochemistry
The mice were exposed to a lethal dose of Ketamine (120 mg/kg) and Xylazine (20 mg/kg) and transcardially perfused with 4% paraformaldehyde (PFA) in PBS. The brains were sectioned with a vibratome (Thermo Scientific Microm HM 650 V) at a thickness of 80 μm. Selected sections were stained with DAPI, mounted with Mowiol, and imaged with an epifluorescent microscope (Leica DM6000).

### Behavior
Behavioral experiments were conducted during the light cycle and experimenters were blinded to the experimental identity of the animals.

**Optogenetic stimulation.** The implanted fiber was coupled to an optical patch cord (Thorlabs GmbH) attached to a laser output module (473 nm) (Shanghai Laser Optics Century Co., Ltd.). The laser power at the fiber tip was measured with a power energy meter (Thorlabs GmbH). Irradiance values for layers 5 and 6 were estimated based on previous measurements in mammalian brain tissue[72]. For 10 mW measured at the fiber tip (fiber NA = 0.39; fiber radius = 100 μm), the irradiance is 318.18 mW/mm², which corresponds to 3.47 mW/mm² at

the level of L5 (0.75 mm cortical depth) and 1.54 mW/mm² at the level of L6 (1 mm cortical depth).

**Laser protocols.** Single laser pulse trials were 5 s long and consisted of continuous laser stimulation. Mechanical and thermal stimuli were applied during laser pulses. Optical stimulation blocks were interspersed with at least 30 s non-stimulation blocks.

**Measurements to determine individual paw lifting laser intensity thresholds.** Laser stimuli (5 s continuous) were applied at 10, 8, 6, 4, 2, 0 mW (318.18 254.55, 190.91, 127.27, 63.64, and 0 mW/mm²) five times at each intensity and paw lifting probabilities were calculated for each intensity (Fig. 1e). For each mouse, we determined the highest intensity resulting in a lack of paw lifting and used this individual laser intensity during sensitivity measurements (mechanical and thermal).

**Inflammatory pain model.** About 20 µl of Complete Freund's adjuvant (CFA) was subcutaneously injected into the left hindpaw under anesthesia using isoflurane. Behavioral experiments with the CFA cohorts (von Frey and CPP) were carried out one day after the CFA injection.

**von Frey test.** Mice were habituated to the von Frey test chamber twice a day for 1 h for three consecutive days without von Frey filament stimulation. Each von Frey test session started with an acclimatization period of 15 min. Mechanical sensitivity was quantified as the probability of paw withdrawal in response to the application of von Frey filaments (Aesthesio Precise Tactile Sensory Evaluator, Ugo Basile S.R.L.) to the plantar surface of the left hindpaw (contralateral to the stimulated HL cortex). Eight filaments (0.04–2.0 g of force) were applied five times each in ascending order, with at least 30 s delay between applications. 100% withdrawal probability was reached when one filament provoked a withdrawal response in all five trials. In this case, measurements were stopped and sensitivity for filaments with greater forces was not tested. Mechanical sensitivity was first measured in the absence of laser stimulation (baseline) and after ~1 h in the homecage, the test was repeated with laser stimulation (5 s per trial) of either S1HL L6-CT or L5 pathways. The mechanical stimulus was applied within 1 s after the onset of the 5 s laser stimulation. Withdrawal responses were only considered during the 5 s laser stimulation period. Another round of tests (baseline and laser) was conducted 1 day after the subcutaneous injection of CFA in the left hindpaw.

**Thermal test.** Thermal sensitivity was tested using the Hargreaves setup (Ugo Basile Inc., Italy) equipped with an infrared heat laser (Model 37370-001, Ugo Basile). The heat laser was aimed at the plantar surface of the left hindpaw and produced radiant heat of increasing intensity. The intensity level was set to 35 and the cut-off time to 20 s to avoid damage to the paw. Three heat stimulation trials were applied alone, and then in the presence of optogenetic laser stimulation (238.64 mW/mm²), with 3 min of recovery time between trials. The paw withdrawal latency was measured per trial.

**Conditioned place preference/aversion (CPP/CPA) test.** The setup for this paradigm consisted of two chambers (each 15 cm × 15 cm) separated by a neutral chamber (8 cm × 8 cm). One chamber contained walls with vertical stripes and a cherry scent, while the other chamber contained walls with horizontal stripes and a honey scent. The paradigm consisted of one (CPP) or five (CPA) baseline sessions followed by two conditioning sessions, each lasting 20 min. Prior to each session, animals were lightly anesthetized with isoflurane and were attached to an optic fiber cable, at which point they were restricted to the neutral chamber using removable wall slides. Once recovered from anesthesia, the wall slides were removed and the session was started. During baseline sessions, no optogenetic stimulus was administered. During the conditioning sessions, optogenetic stimulation (8 mW, 254.55 mW/

mm², 20 Hz) was administered when the animal entered the chamber in which it spent less time (in the case of CPP) or more time (in the case of CPA) in the last baseline session. Pulsed stimulation at 20 Hz was chosen to reduce phototoxic effects during prolonged stimulation required for the paradigm. AnyMaze software (Version 7.1, Stoelting Co., Ireland) was used to track the animals' position and time spent per chamber for every session. To assess performance outcomes, a comparison was made between the time spent in the stimulation-paired chamber during the last conditioning session and the time spent in that same chamber during the last baseline session. Preference indices (PI) were computed using the following formula: (time in paired chamber – time in unpaired chamber)/(time in paired chamber + time in unpaired chamber).

**Statistics and data analysis.** All behavioral data are expressed as the mean ± the standard error, and were analyzed using SPSS (Version 28.0.1.0) and R Studio (Version 4.2.0). Unless stated otherwise, two-way ANOVA for repeated measures with Bonferroni tests for multiple comparisons were used. A $p$ value less than 0.05 was considered to be significant. Microscopy images were edited using Fiji/Image J (Version 1.53c). Schematics and figures were created in Affinity Designer (Version 1.10.6), GraphPad Prism (Version 9.1.1), and Matlab 2022a.

## In vitro slice electrophysiology

**Preparation of acute brain slices.** Ntsr1-Cre mice ($n = 3$) were stereotaxically injected with an inhibitory opsin (AAV1-hSyn-SIO-stGtACR2-FusionRed, $5 \times 10^{11}$ vg/ml, Addgene) see "Virus injection and optical fiber implantation" 6 weeks prior to the recordings. Acute brain slices containing the S1HL region were prepared as follows. Isoflurane-anesthetized mice were decapitated, their brain was removed and trimmed, and the forebrain containing the S1HL region was placed in oxygenated (95% $O_2$ and 5% $CO_2$) ice-cold, cutting solution containing (mM): 125 NaCl, 2.5 KCl, 3 $MgCl_2$, 0.1 $CaCl_2$, 25 glucose, 1.25 $NaH_2PO_4$, 0.4 ascorbic acid, 3 *myo*-inositol, 2 Na-pyruvate, and 25 $NaHCO_3$ (pH 7.4). Then, 170-mm-thick slices were cut with a Leica vibratome (VT1200S) and kept for 20 min in warm (36 °C), oxygenated (ACSF solution that contained (in mM): 125 NaCl, 2.5 KCl, 1 $MgCl_2$, 2 $CaCl_2$, 25 glucose, 1.25 $NaH_2PO_4$, 0.4 ascorbic acid, 3 *myo*-inositol, 2 Na-pyruvate, and 25 $NaHCO_3$ pH 7.4. Then, the brain slices were moved to oxygenated ACSF maintained at room temperature for at least 1 h before performing electrophysiological recordings.

**Optogenetic stimulation and electrophysiological recordings from S1HL L6-CT-stGtACR2 neurons.** After recovery, S1HL-containing slices were placed on an RC-27 chamber (Sutter Instruments) mounted under BX51 upright microscope (Olympus), equipped with DIC and fluorescent capabilities. Slices were maintained at 24 ± 1 °C using a dual TC344B temperature control system (Sutter Instruments). S1HL slices were continuously perfused with oxygenated (95%$O_2$/ 5%$CO_2$) ASCF solution containing (in mM): 125 NaCl, 2.5 KCl, 0.1 $MgCl_2$, 4 $CaCl_2$, 25 glucose, 1.25 $NaH_2PO_4$, 0.4 ascorbic acid, 3 myo-inositol, 2 Na-pyruvate, 25 $NaHCO_3$, pH 7.4, and 315 mOsm. Cells were approached and patched under DIC, using 3.0 ± 0.5 MegaOhm glass pipettes (WPI, Inc), pulled with a PC10 puller (Narishige, Japan). Recording pipettes were filled with a current-clamp internal solution containing (in mM): 125 K-gluconate, 20 KCl, 10 HEPES, 0.5 EGTA, 4 ATP-Magnesium, 0.3 GTP-Sodium, 10 Na-Phosphocreatine, osmolarity: 312 mOsmol; pH 7.2 adjusted with KOH. In all recordings, we used a Multiclamp 700B amplifier (Axon Instruments, Inc) controlled by Clampex 10.1 and Digidata 1440 digitizer (Molecular Devices, Inc). Detection and analysis of current-clamp recordings was done with Clampfit 10.1.

To assess the impact of stGtACR2 activation on L6-CT neuron activity, cells expressing red fluorescence were approached and recorded in either loose cell-attached or in whole-cell current-clamp configuration. Cells displaying spontaneous spikes in cell-attached

mode, were challenged with pulses of blue light (480 nm, 5 s) generated via a CoolLED illumination system (pE-300) controlled by a TTL pulse. In whole-cell current-clamp mode, cells were maintained at near resting potentials (-−70 mV) or at more depolarized potentials (-−40 mV) via direct current injection through the patch pipette. Two experiments were performed in the current-clamp mode. First, to assess the impact of stGtACR2 activation on membrane potential and spontaneous spiking activity, cells were challenged with long (5 s) pulses of blue light (480 nm), similar to the cell-attached experiments described above. Second, to determine if stGtACR2 activation leads to changes in the input resistance of L6 neurons, cells maintained at near resting potentials were stimulated with square pulses of current (500 ms, from −100 to +300 pA, 20 pA steps), and the amplitude of membrane potential changes before, during, and after blue light activation, was determined.

### In vivo electrophysiology

**Anesthetized in vivo electrophysiology.** Mice were anesthetized using urethane (1.4 g/kg, i.p.) and maintained using an oxygen-isoflurane mixture (0.2%). The mice were fixed with ear bars and the skull was leveled. A craniotomy was performed above the recording site, and a well was cemented (Paladur) and filled with isotonic Ringer's solution. Sharpened 64-channel silicon probes (impedance -50 kOhm) (Cambridge Neurotech) were inserted into the S1HL cortex (ML = +1.67 mm, AP = −1.0 mm, DV = −1.4 mm), VPL (ML = +1.8, AP = −1.3, DV = −4.5), or POm (ML = +1.2 mm, AP = −1.7 to −2.4 mm, DV = −3.4 mm, 4 shank probe) using a micromanipulator (Luigs Neumann 3-axis Motor), moving at -2 μm/s. The probes were docked to a connector (ASSY-77) with an adapter (A64-Om32x2 Samtec) that was, in turn, connected to an RHD2164 headstage amplifier chip (Intan technologies). Signals were amplified and digitized at a sampling rate of 30,030 Hz via an RDH2000 Intan evaluation board using USB 2.0 interface. An Intan Talker module (Cambridge Electronic Devices, Cambridge, UK) was for data acquisition with Spike2 (v9.06) software.

**Awake in vivo electrophysiology.** Mice between 8–12 weeks (*n* = 4) were recorded on a cylindrical treadmill consisting of a 15 cm diameter foam roller mounted on a custom-built low friction rotary metal axis, attached to two vertical posts. Ntsr1-Cre and Rbp4-Cre mice were stereotaxically injected with an inhibitory opsin (AAV1-hSyn-SIO-stGtACR2-FusionRed, 5 × 10[11] vg/ml, Addgene) see "Virus injection and optical fiber implantation") 2–3 weeks prior to the recordings. At least a week before recording, a polycarbonate two-winged head plate was cemented onto the skull with dental cement (Super-Bond, Sun Medical Co. LTD). A rubber ring was cemented around the craniotomy to create a small ringer reservoir for the reference electrode. Mice were allowed to recover from the surgery for 2 days and, over the next 4 days, were habituated to the cylindrical treadmill in the Faraday cage. Animals were head-fixed on the treadmill apparatus with the head plate. Habituation sessions lasted for -60 min, during which mice freely walked on the cylindrical treadmill and were fed sweetened condensed milk as a reward. A maximum of 24 h prior to recording, a craniotomy was performed over the injection sites. The rubber well was then covered with silicone elastomer (Kwik-Cast, World Precision Instruments) until the experiment. During the recording session, the protective silicone was removed and an acute silicone optrode (H3, Cambridge Neurotech) was lowered into the S1HL. The activity was recorded using the apparatus and software described in the section "Anesthetized in vivo electrophysiology".

**Mechanical stimulation during anesthetized in vivo electrophysiology.** Target regions were identified by brushing or pressing the hindpaw with a brush or cotton swab and assessment of evoked activity from the Spike2 visual interface readout. Mechanical stimulation was automated by a stimulation protocol prepared in Spike2

through interface hardware (Power1401, Cambridge Electronic Design, Cambridge, UK) through a stepper controller, which initiated a motor (Mercury Step C-663 Stepper Motor Controller, PIMikroMove Version 2.25.2.0). Nine von Frey filaments glued (Pattex Sekundenkleber) to a force sensor (Single Tact miniature force sensor) attached to a motor, so the TTL pulse would apply these von Frey filaments to the paw, and pressure information would be relayed to the Intan board. A total of 9 g (9 × 1 g per filament) pressure was delivered for a 5 s duration every 60 s (based on the protocol in ref. 73).

**Optogenetic stimulation during anesthetized and awake in vivo electrophysiology.** An optical fiber (Thorlabs GmbH, NA = 0.22; radius = 52.5 μm) was positioned -0.5 mm perpendicular above the craniotomy. Laser power densities overlapped with the behavioral experiments (-0.5–27 mW at fiber tip corresponding to 57.72–3105.34 mW/mm² and 0.26–14.16 mW/mm² at the level of L6[72]). Light pulses were initiated automatically by an Omicron Light-Hub2 (Wavelength = 488 nm) with a stimulation protocol prepared in Spike2 through interface hardware (Power1401, Cambridge Electronic Design, Cambridge, UK). Trial repetitions per condition were 31–52 trials for VPL recordings and 32–35 trials for S1 recordings, and 22–26 trials for POm.

**Stimulation protocol during anesthetized and awake in vivo electrophysiology.** Experiments consisted of three stimulus conditions of 5 s duration each: mechanical, mechanical + optogenetic laser, and optogenetic laser. The time interval between mechanical and mechanical + optogenetic laser conditions was 60 s, otherwise, the time interval between conditions was 30 s (Fig. 3a). The stimulation in awake recordings consisted only of 5 s laser stimuli every 15 s.

**Spike sorting.** Voltage data were band-pass filtered upon acquisition (500–5000 Hz). Spike2 data files (.smrx) were converted into binary files. The file conversion consisted in reading the electrophysiology channels in the.smrx file, transforming them back into uint16 values from the 16-bit depth analog-to-digital (ADC), and writing them in the resulting.bin file.

Spike sorting was performed semi-automatically using Matlab-based Kilosort 2.5[74] and resulting clusters curated in Phy2 (https://github.com/cortex-lab/phy). Single units with <0.5% refractory period (1 ms) violations and a baseline spike rate >0.1 Hz were accepted for further analysis. Kilosort 2.5 spike-sorting parameters are summarized in Table 1.

### Classification of putative cell types

**Thalamic units.** VPL units were defined as follows. First, probe location within VPL was confirmed histologically (Supplementary Fig. 7): For each recording, we first identified the dorsolateral range of channels which contained units with significant responses to M or ML. All units within that range were assigned to VPL. POm units were defined by the location of the probe, which was composed of four shanks aligned along the anterior/posterior axis, with the shanks distributed in 250 μm intervals. The anterior shank of the probe was stereotaxically targeted to POm based on the following coordinates: (ML = +1.2, AP = −1.7, DV = −3.4).

**Cortical units.** Silicon-probe recording channels were registered to histological layer borders to assign each unit a cortical depth and layer. Layer borders in the S1HL cortex were estimated histologically based on soma sizes and densities using DAPI signals (examples in Figs. 1, 6). Putative L6-CT and L5 units were isolated based on low latency (<9.5 ms) and low-jitter (<3 ms L5, <2 ms L6-CT) responses to laser light pulses (Supplementary Fig. 2). Putative FS-like units were identified by their peak-to-second trough latency below 215 μs[75], and were removed from the optotagged populations. All other electrophysiological analysis included putative FS and RS units. The identity of optotagged units was validated by plotting these units by depth along the cortical axis,

**Table 1 | Kilosort 2.5 spike-sorting parameters**

| Parameter | Value | Parameter | Value |
|---|---|---|---|
| ops.fs | 3.003003003003003e + 04 | ops.sigmaMask | 30 |
| ops.fshigh | 600 | ops.ThPre | 8 |
| ops.Th | [10 2] | ops.sig | 20 |
| ops.lam | 10 | ops.nblocks | 0 |
| ops.AUCsplit | 0.7 | ops.spkTh | –6 |
| ops.minFR | 1/50 | ops.reorder | 1 |
| ops.momentum | [20 400] | ops.nskip | 35 |
| ops.GPU | 1 | ops.Nfilt | 1024 |
| ops.nfilt_factor | 4 | ops.ntbuff | 64 |
| ops.NT | 64*1024+ ops.ntbuff | ops.whiteningRange | 32 |
| ops.nSkipCov | 25 | ops.scaleproc | 200 |
| ops.nPCs | 6 | ops.useRAM | 0 |

demonstrating that optotagged L5 and L6-CT neurons are located at the expected cortical depths in layers 5 and 6, respectively (Figs. 1, 6).

**Spike train analysis.** After spike-sorting, spike times were aligned to stimulus onsets and segregated into stimulation conditions using custom Matlab 2022a scripts. For cleaner visualization in Figs. 3, 4, peristimulus time histograms (bin size = 20 ms) were smoothed using a Matlab port (https://github.com/iandol/spikes/tree/master/Bars) of the Bayesian Adaptive Regression Splines technique[76] using default parameters except for prior_id = 'POISSON'. All further calculations were done on unsmoothed data.

**Response windows.** We considered spike counts and timing within a time window of 1.5 s after the stimulus onset + 500 ms to account for delayed activation of the mechanical stimulus relative to the command signal and baseline activity was taken as 1.5 s before stimulus onset (Figs. 3a, 4a). Laser-only trials were also sampled during the same windows to discard transient effects at stimulus onset (e.g., PSTH peak in Fig. 4d, L6) and to enable comparison between L and ML conditions.

**Unit modulation.** Units were considered modulated in a given stimulus condition if a significant difference was measured between baseline and stimulus-evoked spiking either in terms of absolute spike counts ($p \leq 0.05$, signed-rank test for paired baseline and stimulus-evoked trials) or spike timing detected by using the ZETA test[77].

**Response parameters.** We compared stimulus-evoked changes in single-unit activity across L, M, and ML conditions using spike counts and interspike interval (ISI) statistics to capture both overall spike output and spike timing. The mean spiking rate (r̄) was calculated as the mean spike count per trial, divided by the response window duration of 1.5 s. Response probability (RP) was calculated as (trial count with at least one spike)/total trial count, measured within the 1.5 s response window.

To calculate burst probability (BP) for single units, spikes preceded by an interspike interval of less than 5 ms were considered part of a burst. BP was then calculated as total burst events/(total burst events + total single spike events). Trends reported were relatively insensitive to a burst cut-off ISI of up to 10 ms. Modulation index (MI) was calculated as $MI = (\bar{r}_{stimulus} - \bar{r}_{baseline})/(\bar{r}_{stimulus} + \bar{r}_{baseline})$ as in ref. 55. Data were presented as (first quartile, median, and third quartile) or mean ± SEM.

**Statistical analysis of spike train data.** All statistical analysis was done in Matlab 2022a or R, using built-in or custom-written functions. Unless otherwise stated, data were analyzed by two-way repeated measures ANOVA with a Bonferroni test. See Supplementary Table 1 for exact statistical tests and test outputs (F- and p-values). Paired MI and r̄ data across conditions (per region) were analyzed using the Friedman test, followed by a Wilcoxon signed-rank test. MI and r̄ comparisons across regions (Supplementary Table 2) was done with a mixed-model ANOVA followed by a rank-sum test for pairwise differences. Comparisons of the proportion of either positively, negatively, or unmodulated units per stimulation condition were made by a $X^2$ proportions test, followed by the Marascuillo procedure for multiple comparisons. Statistical differences in the proportions of responsive units across conditions were assessed using McNemar's test in the case of paired data or a two-proportions $X^2$ test in the case of unpaired data (custom written), e.g., between cortical layers.

**Reporting summary**
Further information on research design is available in the Nature Portfolio Reporting Summary linked to this article.

## Data availability
All the data are in the manuscript or in the supplementary material. Source data is provided as a Source Data file with this paper and under the following link: https://doi.org/10.11588/data/D2O0JZ[78]. Source data are provided with this paper.

## Code availability
Code necessary to reproduce the Matlab-generated figures in this study are provided and maintained at https://github.com/rebecca-mease/Ziegler_et_al_2023[79].

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

## Acknowledgements

The authors are grateful for technical assistance from Manfred Oswald and Zheng Gan, and software development and support from Liam Keegan (Scientific Software Center, Heidelberg University). Funding was provided by the Deutsche Forschungsgemeinschaft (DFG): Collaborative Research Center 1158 (A.G. and R.A.M.: project B10; R.K.: project B01, B06; T.K.: project B08, C.A.: project S02), DFG individual grant (GR3757/3-1, A.G., E.I.-C., J.M.-C.), the Chica and Heinz Schaller Stiftung and the Brigitte-Schlieben-Lange Programm by the Ministry of Science, Research and the Arts Baden-Württemberg (R.A.M.). C.A. is also supported by the Chica and Heinz Schaller Stiftung, the NARSAD Young Investigator award 2019, and a 2021 Fritz Thyssen grant. The authors acknowledge support by the state of Baden-Württemberg through bwHPC and the German Research Foundation (DFG) through grant no INST 39/963-1 FUGG (bwForCluster NEMO) and the data storage service SDS@hd supported by the Ministry of Science, Research and the Arts Baden-Württemberg (MWK) and the German Research Foundation (DFG) through grant INST 35/1503-1 FUGG. For the publication fee, we acknowledge financial support by Deutsche Forschungsgemeinschaft within the funding program "Open Access Publikationskosten" as well as by Heidelberg University.

## Author contributions

Conceptualization and project administration: A.G. and R.A.M., Methodology: S.A.-G., J.M.-C., L.L.T., S.K., R.K., and T.K., Investigation: K.Z., R.F., A.J.G., J.B., and C.A., Data analysis and analysis tools: K.Z., R.F., A.J.G., J.B., E.I.-C., R.A.M., and C.A., Visualization: K.Z., J.B., R.F., A.J.G R.A.M., and A.G., Funding acquisition and supervision: A.G., R.A.M., R.K., and T.K., Writing—review and editing: K.Z., R.F., A.J.G., J.B., E.I.-C., J.M.-C., R.K., T.K., C.A., R.A.M., and A.G.

## Funding

## Competing interests

The authors declare no competing interests.
