## [Peer Review File · Nature Communications]

Primary Somatosensory Cortex Bidirectionally Modulates Sensory Gain and Nociceptive Behavior in a Layer-Specific MannerREVIEWER COMMENTS

Reviewer #1 (Remarks to the Author):

In the paper entitled “Primary Somatosensory Cortex Bidirectionally Modulates Sensory Gain and Nociceptive Behavior in a Layer – Specific Manner” by Ziegler et al., suggested Layer 6 (L6) neuron in primary somatosensory cortex (S1) drive aversive hypersensitivity and spontaneous nociceptive behavior that associated with strong suppression of L5 neurons.

The purpose of this research is potentially important in this research field. The manuscript is well written enough, however the causality of the L5 neuron to contribute on pain like behavior is not enough and several points need to be addressed.

Major)

1. In this research, most of the data to show L6 neuron to contribute on the pain like behavior is due to their activation and not inhibition. Author should perform inhibition experiment for L6 neuron to see mice behavior and the activity of the L2/3, L4 and L5 neurons.
2. Author need to quantify the function connection of L6 neurons to the other layer of the neurons.
3. For Figure 2, I wonder if the laser induced pain stimulation showed sustain effect for the mice to enter the room, like we found in fear conditioning. What is the time course?
4. Author need to manipulate the activity of L5 neurons with the activation of L6 neurons to see the behavior output
5. For the pain model mice like in CFA mice, what about the L6 neuronal population to be activated. Author need to test using c-Fos staining and so on.
6. If laser light influences the mouse behavior in aversion test as author mentioned, author needs to control light to stimulate visual input.

(Minor)

- 1) Author need to check the neuronal subtype which expressed the ChR2.
- 2) Mixed upper- and lower-case letters, such as von Frey test and isoflurane. Please unify them.
- 3) Change the description of supplementary figure from Fig. S5x to Supplementary Fig. 5x.
- 4) In Fig.6a, part of letters is missing.
- 5) “Complete Freund's adjuvant” on page 30 is spelled differently.
- 6) Author needs to indicate the test statistic (e.g. F, t, r) with confidence intervals, effec

Reviewer #2 (Remarks to the Author):

The roles of primary somatosensory cortex (S1) in innocuous and noxious signals processing have been well documented, but the roles of specific layers in pain signals processing is still remaining elusive. Using state-of-the-art neuroscience technology, the author characterizes specific roles of L5 and L6 in nociception and pain. The overall claim is that activation of L6 neurons can drive aversive

hypersensitivity and spontaneous nocifensive behavior, while activation of L5 neurons reduced sensory sensitivity and normalized inflammatory allodynia. Meanwhile, activation of L6 can enhance or suppress neuronal response of the somatosensory thalamus and L5 of S1, respectively. The main finding of the work is interesting, the quality of data is good and results are convincing. Nevertheless, I have an issue with the scope of the work.

Major

1. The manuscript is kind of long and tedious, not easy to get points. Should be shorted and tightened. Also, it is need to tighten the rationale and clarify the potential impact of this study.
2. By only examining hindpaw withdrawal probabilities in response to mechanical stimulation during optical activation of L6-CT neurons, the conclusion that “L6-CT activation induced mechanical hyperalgesia and allodynia in naive animals, and exacerbated these effects in an inflammatory model” is not convincing. Whether the activity of L6-CT neurons was enhanced in the model mice? I am also curious that if optical inhibition of L6-CT neurons induces analgesic effect and conditioned place preference?
3. L6-CT neurons are targeted and manipulated using Ntsr1-Cre mouse line, and the authors showed optical activation of L6-CT neurons enhance or suppress neuronal activity of VPL and L5, respectively. Are these opposite effects caused by direct projections from L6-CT neurons? If these opposite effects are caused by direct projection from L6-CT neurons, whether these Ntsr1 neurons innervate L5 and VPL by different neurotransmitters? In addition, L2/3 and L4 neurons showed increased and decreased activity in response to L6-CT neuronal activation. Do Ntsr1 L6-CT neurons release different neurotransmitters in a region specific manner?
4. For fig.3, “L6-CT activation enhances VPL output to cortex”. The data in the figure showed either L6-CT neuronal activation or mechanical paw stimulation enhance the activity of VPL neurons. How can they conclude VPL outputs to cortex are enhanced without further evidence.
5. “Both L2/3 and L4 showed robust increased spiking in response to paw stimulation, in both proportions of sensory-encoding units (76% and 61%, for L2/3 and L4, respectively; Fig. 4c) and firing rate modulation (Fig. 4d). In contrast to this relative uniformity of sensory encoding, the effect of L6-CT activation was heterogenous and relatively weak, as some units 25 were enhanced or (suppressed) by the laser alone (L): L2/3: 31% (46%), L4: 40% (43%).” Since the L6 neurons projects to somatosensory thalamus, which reciprocally connects with different layers of somatosensory cortex. It is thus unsurprising that the recorded responses are heterogenous. Blockade of the projection from the VPL to the S1 is preferred to examine the effect of L6-CT neuronal activation on L2/3 and L4 neurons.
6. In the experiments examining the role of L5-PT neurons in pain processing, the conditioned place preference test rather than CPA should be employed to examine whether L5-PT neuronal activation induce preference since previous data showed activation of these neurons decreased mechanical sensitivity in CFA-injected mice. Besides, analyses of the preference indices show that laser stimulation in S1HL in L5-EGFP controls caused comparable avoidance for the laser paired chamber. The authors

explained this avoidance may arise entirely from visual avoidance to the laser flashes. However, this avoidance was not observed in L6-EGFP mice which treated with similar light. Therefore, these results need new explanation.

7. In vivo recording showed that optical activation of L6 neurons suppress L5 neurons and activate VPL neurons separately. Behaviorally, L6 neuronal activation induced spontaneous nocifensive behavior and L5 neuronal activation normalized inflammatory allodynia. The reciprocal connections between L5, L6 and VPL in pain state is not clear.

Reviewer #3 (Remarks to the Author):

Ziegler et al. have manipulated cortical activity in the hindlimb primary somatosensory cortex (S1) using cell-type specific mouse driver lines to study the functional impact on the perception of innocuous and noxious sensory signals. Using a combination of optogenetics and in vivo electrophysiology, the authors report that in awake mice, activation of L6 CT cells using the Ntsr1-Cre driver line results in nocifensive behaviors and increased paw sensitivity, as well as exacerbates inflammatory allodynia and elicits an aversive experience. The authors then demonstrate that the pronociceptive effect of L6 CT activation involves enhancing the excitability and sensory throughput of the ventral posterolateral thalamus (VPL) to mechanical paw stimulation in anesthetized mice. In the cortex, the authors found that L6 stimulation completely suppressed L5 activity but had only a modest influence on the overall spiking responses in L4 and L2/3, despite the overall elevated activity in VPL. This observation is largely consistent with the idea that L6 serves as an intracortical gain control (Olsen et al., 2012, Nature). Given L6 stimulation induced both nocifensive behavior and paw hypersensitivity while also eliminating L5 activity, the authors next probed the role of L5 pyramidal tract (PT) neuron activity in nocifensive behaviors and modulating paw sensitivity using the Rbp4-Cre driver line. They report that L5 PT activity in S1 has opposing effects to L6 CT activity, reducing mechanical sensitivity and normalizing inflammatory hypersensitivity.

This study is technically impressive, with some interesting results. The observation that optical stimulation of L6 CT cells can influence thalamic throughput and cortical sensitivity is not novel. Still, as the authors nicely highlight in their manuscript, the role of CT cells in maladaptive hypersensitivity, particularly involving pain perception, has received very little attention. Thus, their finding that L6 CT modulates sensory gain and nociceptive behavior is important to the field. Furthermore, the demonstration that enhancing L5 activity has an opposite effect (or anti-nociceptive function) is also new and potentially interesting. To come to these conclusions, the authors combined in vivo electrophysiological and optical techniques as well as different behavioral approaches in a nicely integrated way. The paper is well written and illustrated, for the most part.

In the manuscript, however, some critical loose ends need to be addressed. Most importantly, the neural circuit mechanism underlying the reported layer-specific bidirectional control of nociceptive

behavior.

My questions and concerns are listed in roughly decreasing order of importance:

1. My main concern has to do with the mechanism underlying the bidirectional modulation of sensory gain and nociceptive behavior. In their first set of experiments, the authors suggest that L6 CT enhancement of nocifensive behaviors and increased paw sensitivity is a consequence of enhanced thalamic throughput and, in parallel, strong suppression of L5 neurons. However, it is unclear how much of the behavioral modulation is due to elevated thalamic activity versus suppression of L5 activity. They sort of address this in their second set of experiments, where they show that elevated L5 activity reduces sensory sensitivity, but again the mechanism underlying how elevated L5 activity decreases sensitivity is unclear (not even investigated). This is critical since the antinociceptive actions of elevated L5 activity are unlikely to involve VPL, at least directly, since L5 cells don't project to VPL.

2. Related to point 1 above, it would be interesting to know more about the bi-directional gain control of each layer in isolation, as well as how L5 and L6 interact. Specifically, what is the impact of L5 activation/suppression on L6 activity? Does L5 suppression alone cause hypersensitivity? What happens when L6 activity is suppressed? Does sensitivity decrease? Does L5 activity increase?

3. The authors also don't consider another feature of the circuit: the higher-order posterior medial thalamus (POm). They present data about L6 modulatory inputs to VPL, but never report on the influence on the POm. This seems critical since this thalamic region is thought to serve higher-order functions, and it receives input directly from L5 and L6 cells.

4. Throughout the manuscript, we are led to believe that the Rbp4-Cre line is specific to L5 PT cells, but we are provided no evidence that this is the case for S1 HL. Based on the original report and subsequent papers (as well as the mouse connectivity map from Allen Institute), I believe this mouse line expresses Cre in both PT and IT cells (Gerfen et al., 2013, Neuron). Without such new data, the author's conclusions about L5 PT cell involvement must be revised.

5. Additional information regarding the delay between light stimulation and behavior would be helpful for Figures 1 and 5.

6. Identification of opsin-expressing cells should be described in more detail in Figure 1 and 5. Plots showing the criteria used to distinguish optotagged L5/L6 cells from non-optotagged cells and how FS-like neurons were separated from optotagged cells.

Minor

1. Page 4, line 1: should be "and their corticothalamic axons/terminals".
2. Figure 1d: the values on the y-axis are small and hard to see.
3. The viral titer and expression time should be reported

We thank the reviewers for appreciating our study and that they find that *“The main finding of the work is interesting, the quality of data is good and results are convincing.”, that “The manuscript is well written” and that “This study is technically impressive, with some interesting results.”*. We appreciate the constructive comments that helped us to sharpen the presentation of our results and to include new key data into the revised manuscript.

We addressed all concerns and critical points by performing several sets of new behavioral and electrophysiological experiments and analysis, most critically including inhibition of L5 neurons to strengthen the causal relationship between cell-type-specific neuronal activity and behavioral phenotypes. The manuscript has been thoroughly revised and we are delighted to report that all new data support the main conclusions and impact of the study and provide further insight into the underlying circuitry.

As a result of these revisions, we have largely reworked Figs. 2-5: Fig. 5 became Fig. 6, we added two new main figures (Figs. 5, 8), and added seven new Supplementary Figures and a new Supplementary table. In keeping with these changes, we streamlined the results text for Fig. 4 to now indicate the effects of L6-CT activation with respect to the layer-specific population responses to L/M/ML conditions shown in the main figure, as the previous percentage values referenced material now moved to Supplementary Fig. 8.

We provide a point-by-point reply to the reviewers below.

REVIEWER COMMENTS

Reviewer #1 (Remarks to the Author):

In the paper entitled “Primary Somatosensory Cortex Bidirectionally Modulates Sensory Gain and Nociceptive Behavior in a Layer – Specific Manner” by Ziegler et al., suggested Layer 6 (L6) neuron in primary somatosensory cortex (S1) drive aversive hypersensitivity and spontaneous nociceptive behavior that associated with strong suppression of L5 neurons.

The purpose of this research is potentially important in this research field. The manuscript is well written enough, however the causality of the L5 neuron to contribute on pain like behavior is not enough and several points need to be addressed.

Major)

Response: We thank the reviewer for pointing out the importance of this study and the constructive suggestions for changes, which we have implemented as described below. Since the causal link between L5 neurons and pain-like behavior is brought up by the reviewer in the general statement, we would like to address it here. We performed several additional investigations in which the effect of activating or suppressing L5 activity was assessed on the behavioral and electrophysiological level. This new data shows that suppression of L5 activity causes increased sensitivity (new Fig. 5), thereby complementing the stimulation experiments in the original paper (now Fig. 6), which showed the opposite: decreased mechanical sensitivity and normalization of inflammatory allodynia. Most importantly, we show in new experiments that L5 suppression causes aversion while L5 stimulation in the inflammation model causes a place preference (Fig. 5 and 6, respectively), both observations are new key results to further support the antinociceptive potential of L5 interventions. Furthermore, on the circuit level, L5 suppression increases L6-CT firing (new Fig. 5). We thank the reviewer for

stimulating these additional investigations as these new data strengthen the causality between L5 activation and pain relief and provide further insight into the interactions between L6-CT and L5, as we describe in more detail in the respective point-by-point replies below.

1. In this research, most of the data to show L6 neuron to contribute on the pain like behavior is due to their activation and not inhibition. Author should perform inhibition experiment for L6 neuron to see mice behavior and the activity of the L2/3, L4 and L5 neurons.

Response: We appreciate the reviewers interest in the effects of L6 inactivation on behavior and S1 activity during nociception. We implemented the suggested experiments in a new set of behavioral and *in vitro* electrophysiological investigations in which we optogenetically inhibited L6 activity as suggested. We introduced the inhibitory opsin stGtACR2 (stGtACR2) into our study, which is a soma-targeted anion-conducting channelrhodopsin (Mahn et al., 2018). After optimizing soma-restricted expression in the Cre lines in S1HL cortex, we characterized the efficiency of stGtACR2 to suppress neuronal spiking *in vitro*. stGtACR2-expressing L6-CT neurons were strongly hyperpolarized in response to blue light stimulation and spiking was efficiently suppressed (new Supplementary Fig. 10).

We then assessed the effect of L6-CT suppression on nociception and aversion. In brief, we found that mechanical sensitivity was only slightly increased ($n = 10$, $p = 0.004$), while heat sensitivity remained unchanged ($n = 11$, $p = 0.88$). Furthermore, new real-time CPA and CPP experiments showed that L6-CT suppression does not cause aversion or causes limited preference that is only observable in the between-group analysis of PIs. Thus, while L6-CT activation has drastic pronociceptive and aversive effects (Figs. 1, 2), L6-CT suppression has comparably little effects on the behavior (new Supplementary Fig. 11). These mild effects are consistent with the reported mild effects of L6-CT suppression but strong effects of L6-CT activation on cortical and thalamic activity, found in the whisker thalamocortical system studying the same neuronal subtype (Pauzin & Krieger, 2018).

In addressing the last part of the question of how L6-CT suppression affects L2/3, L4 and L5 neurons, we encountered a technical limitation due to a floor effect of suppressing L6CT. L6-CT activity was low under our experimental conditions (median spike rate = 0.0075 Hz, IQR = 0.0188 $n = 75$ units, 3 mice, data from Fig. 4). These spike rates could not be further suppressed in our *in vivo* conditions by optogenetic suppression, even though the opsin works very efficiently as we show *in vitro* (Supplementary Fig. 10). Low average L6 activity has been reported before (de Kock & Sakmann, 2009; Vélez-Fort et al., 2014) and it is currently not clear what drives L6-CT activity under physiological conditions. Thus optogenetic stimulation of L6-CT likely evokes strong activity compared to the low baseline rates (although see our quantification in Supplementary Fig. 8 that shows that even at the highest laser intensities L6-CT neurons fire at around 8Hz, which is well within the range of firing rates of 10 Hz reported for L6 neurons in the whisker system (O'Connor et al., 2010). Thus, the conclusion that L6-CT activation causes hypersensitivity is valid, even though it is currently not known what drives L6-CT neurons under physiological or pathological conditions. Please see also revised discussion (starting at line 662). Importantly, our observation that L6-CT activity can cause adverse sensory experiences points in the direction that the activity of L6-CT neurons must be tightly regulated and that high sensory gain is not necessarily beneficial.

Nevertheless, we fully acknowledge the reviewer's main point that the effect of L6-CT neurons onto other S1 neurons was difficult to get in the original manuscript. We therefore added summary plots to Fig. 4 (f, g) and Fig. 8 which show the net action of L6-CT on the spontaneous and evoked activity in the cortical column: enhancement of L2/3 and 4 and suppression in L5. Please also see our reply to the next comment.

The L6-CT suppression results strongly contrast with the results obtained from a new set of matching L5 suppression experiments. Here, L5 suppression increased mechanical sensitivity ($n = 6$; $p = 0.014$), had no effect in the CPP paradigm but instead caused aversion in the CPA paradigm – all of these observations are in support of the antinociceptive function of L5 neurons that our study suggests (Figs. 5, 6). Notably, we also provide new data that shows that L5-ChR2 stimulation causes a strong place preference in CFA animals, suggesting that this antinociceptive function of the S1HL L5 pathway indeed alleviates pain in a preexisting pain condition (new Fig. 6 h-j). We thank the reviewer for pointing us in the direction of suppressing cortical output pathways which give further support for the bidirectional modulation of nociception by S1HL output pathways, emphasize the antinociceptive function of the S1 L5 pathway, and provide further insight into their interactions. We compiled a summary of the outcomes of the main behavioral manipulations in new Supplementary table 3.

2. Author need to quantify the function connection of L6 neurons to the other layer of the neurons.

Response: We agree that the effects of L6-CT activation on other cortical layers and to the thalamus is a very important point and that we did not clearly enough present how our results compare to the known connectivity of L6-CT in other cortical areas. We addressed this point in the following ways. As suggested, we included a new and concise presentation of L6 effects onto the other layers in S1HL (layer-by-layer comparison of modulation indices across conditions, Fig. 4f) which summarizes the net effects of L6-CT onto the spontaneous and evoked activity in the cortical column: enhancement of L2/3 and 4 and suppression in L5. In addition, we summarized the results from the thalamic and cortical data (Figs. 3 and 4) in a new compact graphic (Fig. 4g). We also include a new schematic summarizing the functional and behavioral ramifications of the present work (Fig. 8). We further revised the manuscript at several places—notably (starting at lines 215; 353, 596, 633) to emphasize the functional effects of L6-CT on S1HL layers and the thalamus. We recapitulate the key points here:

- 1) Previous studies have shown that L6-CT neurons can both suppress and enhance downstream neurons in the cortex and in the thalamus (Crandall et al., 2015; Frandolig et al., 2019; Mease et al., 2014; Olsen et al., 2012; Pausin & Krieger, 2018). L6-CT neurons are glutamatergic and they form functional connections with inhibitory and excitatory neurons: in the thalamus, L6-CT connects to thalamic relay neurons and to inhibitory neurons in the TRN, which in turn inhibit relay neurons. Similarly, in the cortex, L6-CT neurons connect to excitatory and inhibitory cortical neurons. Summarizing the existing literature on L6-CT, the overall net effect of L6-CT stimulation on a coarse level is suppression of cortex and enhancement of thalamic relay neurons (Mease et al., 2014; Olsen et al., 2012; Pausin & Krieger, 2018).
- 2) However, the effects of L6-CT depend on other factors such as the statistics of the L6-CT activity (Crandall et al., 2015) and on the specific cell-types (Pausin & Krieger,

2018). Indeed, our analysis in Fig. 3 and 4 show that the degree of suppression and enhancement of neuronal responses by L6-CT was highly cell-type-specific. As a likely mechanism underlying the L6-CT-mediated aversive hypersensitivity, we observe enhancement of the VPL, L4, L2/3 axis, consistent with an enhanced sensory transmission through the canonical circuitry of the lemniscal pathway of the lateral pain system, which carries body signals to S1 (Groh et al., 2017). In contrast, the functional connection of L6-CT to L5 was strongly inhibitory (Fig. 4). Together, this suggests that at the same time as L6-CT stimulation boosts sensory transmission, L6-CT neurons suppress antinociceptive L5 output from the cortex (which we support with several new data sets), all together resulting in abnormal hypersensitivity, nocifensive behavior, and aversion.

3. For Figure 2, I wonder if the laser induced pain stimulation showed sustain effect for the mice to enter the room, like we found in fear conditioning. What is the time course?

Response: Thank you for your comment. To assess whether the observed aversion of L6-CT stimulation is sustained, we analyzed changes in the number of entries into the paired chamber over time within each conditioning session. We found that (1.) re-entries drastically dropped over the time course of the first conditioning session in experimental animals (which did not happen in control animals), and (2.) that in experimental but not control animals, this effect was sustained already at the beginning and throughout the entire course of the second conditioning session. These results strongly indicate learned aversion (in addition to real-time aversion). In conclusion, we can confirm that animals avoid the stimulated chamber even if the laser is off, due to previous exposure and learned association between chamber and laser. We added this new data as Supplementary Fig. 6 and reference the finding of the persistent effect in the results (lines 175, 204). We thank the reviewer for bringing up this excellent point, which strengthens the finding L6-CT activation is strongly aversive.

4. Author need to manipulate the activity of L5 neurons with the activation of L6 neurons to see the behavior output

Response: We implemented this suggestion by conducting new experiments in which we aimed to express ChR2 in both L5 and L6 neurons and quantified the mechanical sensitivity using the von Frey test. Layer-specific expression in wild-type mice is by nature less specific than using Cre-driver lines (the approach that allowed our central cell-type-specific conclusions in the main paper). To address the reviewer's suggestion we tried the best we could to restrict ChR2-expression to layers 5 and 6 while preventing leakage to upper layers of S1 (please see panel a below). Optogenetic stimulation in these mice led to a small but significantly decreased sensitivity (see panel b caption below). The fact that simultaneous activation of L5 and L6 neurons resulted in a behavioral phenotype more closely resembling that of our previous L5-specific stimulation (Fig. 6) suggests that the antinociceptive effect of L5 dominates over the L6 effect when both are stimulated simultaneously. Our interpretation is that the effect is mediated via the L5 long-range connections to subcortical circuits (Frezel et al., 2020), in which L5 inputs reduce ascending sensory signals and/or activate descending modulation before these signals even reaches the L6-CT circuitry, which is restricted to the thalamocortical system. One alternative interpretation on technical grounds is that the dual stimulation of L5 and L6 preferentially results in stronger activation of L5 due to less light reaching L6. If the reviewer agrees, we would leave this data in the response letter as the scope of the paper is already quite large (see also comment by reviewer 2) and we prefer to

center the conclusions of the paper on data obtained from cre-drive lines.

(a) Expression of ChR2-EYFP (green) in layers 5 and 6 of S1HL cortex of wildtype littermates (Ntsr1-Cre negative); layer borders (dashed lines) estimated based on soma sizes and densities using DAPI signals (blue). Virus cocktail 1:1 AAV1 Cre (AAV1-hSyn-Cre-WPRE-hGH, 1.8×10^{13} vg/ml, Addgene) + DIO ChR2 EYFP (AAV2-EF1a DIO-ChR2(H134R)-EYFP-WPRE-pA, 5.7×10^{12} vg/ml, Zürich vector core).

(b) Within-animal comparison of paw withdrawal probabilities in response to graded von Frey stimulation of the left hindpaw with (blue, Laser on, 5 s continuous pulse) and without (black, Laser off). L6-CT stimulation in S1HL of layers 5 and 6 ChR2-EYFP animals ($n = 5$ mice, $p = 0.049$).

5. For the pain model mice like in CFA mice, what about the L6 neuronal population to be activated. Author need to test using c-Fos staining and so on.

Response: We appreciated the hypothesis of the reviewer that L6 activity might be increased in the CFA model. We addressed this question in two new sets of experiments in the CFA model. First, we recorded the activity across layers of the hindlimb primary somatosensory cortex, contralateral to a hindpaw injected with either CFA ($n = 3$ mice) or Saline as a control ($n = 3$ mice) 24 hours prior to experiment. When comparing the distribution of baseline firing rates among opto-tagged L6 units, we saw a significant increase in firing rate between the CFA L6 and Saline L6 (170 optotagged units from a total of 1238 S1 units). The median increase in firing rate in L6 in the CFA model was 0.016 Hz (Saline: 0.0075 Hz, CFA: 0.0237 Hz; $p < 0.01$, Wilcoxon Rank Sum). Secondly, as the reviewer suggested, we conducted a c-FOS survey in unilaterally injected CFA animals ($n = 2$) and quantified c-Fos levels in S1 cortices in both hemispheres. Consistent with the electrophysiological study, we observed increased numbers of c-Fos-positive L6 neurons in the hemisphere, contralateral to the CFA-injected paw; however the differences in c-Fos levels did not reach significance levels ($p > 0.05$, one-way ANOVA). In summary, we found a small but significant effect of CFA on the firing rates of L6-CT neurons which did not reach significance in the less sensitive c-Fos survey. Increased baseline activity in S1 is in accordance with previous studies of the effect of persistent pain on neuronal firing rates and c-Fos levels (Chang et al., 2008; Quiton et al., 2010).

We would like to underscore that the hypothesis that inflammatory states are associated with or even caused by L6 hyperactivity is indeed interesting, but that none of our results or

conclusions rely on knowing whether CFA increases L6-CT activity. We added and discuss literature on the relation between pain states and increased S1 neuronal activity (line 661). In the interest of keeping the paper concise (see also comment by reviewer 2), we would prefer to provide this additional data as reviewer attachment, provided the reviewers and editor agree:

(a) Spontaneous firing rates of optotagged L6 units of mice injected with either saline or CFA in the contralateral hindlimb (n = 3 per group).
 (b) Total number of cFos-positive L6 cells in S1HL cortex slices (2 animals, 4 slices per animal) contralateral and ipsilateral to the CFA-injected hindlimb.
 (c) Percentage of all cFos-positive cells in S1HL cortex slices found in L6 (2 animals, 4 slices per animal) contralateral and ipsilateral to the CFA-injected hindlimb.

6. If laser light influences the mouse behavior in aversion test as author mentioned, author needs to control light to stimulate visual input.

Response: This is an important point, which we can clarify. We controlled leakage of the laser light by coating the fiber implant in black cement. This information was missing in the methods and is now added (see “Virus injection and optical fiber implantation”, line 715). The shielding effectively reduced the leakage of the laser, however even if the shielding would entirely prevent any light leaving the brain, dispersion in the tissue is unavoidable and some light reaches the retina from within the brain, an unavoidable side-effect of optogenetic implants. We rigorously controlled for the effects of visual stimulation with control cohorts expressing EGFP in L5 or L6-CT which were exposed to the same conditions as the experimental animals, a control strategy established previously (Tan et al., 2017, 2019). Indeed these controls showed that laser alone causes a measurable aversion, which was consistently observed in all control cohorts (Figs. 2h, Supplementary Fig. 11d (L6-EGFP), Fig. 5g, Supplementary Fig. 9b (L5-EGFP)). With these control measures we were able to dissect the effect of neuronal stimulation from the effect of laser alone, which showed that L6-CT, but not L5 stimulation causes aversion (best seen when comparing PI values and statistics of the experimental and control cohorts (Figs. 2h, Supplementary Fig. 9b)). Thus, we can confirm that laser alone has a behavioral effect but that we thoroughly controlled for this effect by the study design. We thank the reviewer for bringing this important point, which we clarified in the revised manuscript as noted above and in the following portions of the Results (starting at line 170) and Supplementary (line 120):

“The preference index for the paired chamber (PI) also dropped in L6-EGFP mice, indicating that the laser itself is optically aversive. However, the laser effect was significantly stronger in L6-ChR2 mice (Fig. 2h), who developed an immediate and persistent avoidance of the stimulated chamber not observed in L6-EGFP controls (Supplementary Fig. 6).”

Supplementary Fig.9: “Repeating the real-time place aversion paradigm (CPA) from Fig. 2 but with L5 stimulation, shows that L5-ChR2 mice spent less time in the laser-paired chamber relative to the time spent in the same chamber during the baseline session (i.e. without optogenetic stimulation). However, the avoidance in L5-ChR2 animals was much less pronounced compared to L6-ChR2 animals (Fig. 2 h-j). Furthermore, the chamber preference index shows that this avoidance effect is indistinguishable between L5-ChR2 and L5-EGFP controls (Supplementary Fig. 9b) suggesting that the avoidance stems entirely from the laser light (as seen also in the L6-EGFP controls, Fig. 2j). We conclude that L5 activation is much less aversive, if at all, compared to L6-CT activation. “

(Minor)

1) Author need to check the neuronal subtype which expressed the ChR2.

Response: We can clarify this point. The neuronal subtypes of these BAC-driver lines from the Gensat Project have been characterized in detail in previous studies and the two lines can be regarded as the most widely used Cre-lines to specifically target L5 and L6-CT neurons. The L5 line (Rbp4_KL100) is a pan layer 5 line, displaying cre-expression restricted to most layer 5 neurons (PT and IT) throughout neocortical and periallocortical areas (Gerfen et al., 2013). This line has been frequently used to specifically target opsins to L5 neurons; a few examples are (Beltramo et al., 2013; Guo et al., 2020; Kirchgessner et al., 2021; Onodera & Kato, 2022; Prasad et al., 2020; Qi et al., 2022).

The L6 line (Ntsr1cre GN220) is specific for the corticothalamic (CT) subtype in layer 6 of neocortex (Gong et al., 2007). The Ntsr1-Cre line selectively targets those neurons whose axons innervate both cortex and thalamus (L6-CT) and not those whose axons remain within the cortex (Bortone et al., 2014). L6-CT neurons' subcortical projections are restricted to the thalamus (TRN, first-order and higher-order nuclei) but L6-CT axons also from local branches in the cortex (Crandall et al., 2017), in which they engage excitatory and inhibitory neurons (Frändolig et al., 2019; Olsen et al., 2012). The Ntsr1-Cre line has been frequently used to specifically target opsins to L6-CT neurons; some examples are (Clayton et al., 2021; Kirchgessner et al., 2021; Mease et al., 2014; Spacek et al., 2022).

Because the neuronal subtypes of these lines have been well-established (including in the S1HL cortex (Guo et al., 2020), we choose these two lines to differentially target the two major cortical output pathways for the first time in the S1 cortex in the context of nociception.

2) Mixed upper- and lower-case letters, such as von Frey test and isoflurane. Please unify them.

Response: Thank you; we standardized “von Frey” and “isoflurane”.

3) Change the description of supplementary figure from Fig. S5x to Supplementary Fig. 5x.

Response: Thank you, we corrected this.

4) In Fig.6a, part of letters is missing.

Response: Thank you, we changed this.

5) “Complete Freund's adjuvant” on page 30 is spelled differently.

Response: Thank you, we changed this.

6) Author needs to indicate the test statistic (e.g. F, t, r) with confidence intervals, effect

Response: Thank you, we have added more test statistics (F, P values, 95% confidence interval) for all experiments to the Supplementary table 1. Since we did not do regressions or report correlation coefficients in this paper we did not add r-values.

Reviewer #2 (Remarks to the Author):

The roles of primary somatosensory cortex (S1) in innocuous and noxious signals processing have been well documented, but the roles of specific layers in pain signals processing is still remaining elusive. Using state-of-the-art neuroscience technology, the author characterizes specific roles of L5 and L6 in nociception and pain. The overall claim is that activation of L6 neurons can drive aversive hypersensitivity and spontaneous nocifensive behavior, while activation of L5 neurons reduced sensory sensitivity and normalized inflammatory allodynia. Meanwhile, activation of L6 can enhance or suppress neuronal response of the somatosensory thalamus and L5 of S1, respectively. The main finding of the work is interesting, the quality of data is good and results are convincing. Nevertheless, I have an issue with the scope of the work.

Response: We thank the reviewer for the in-depth review and we are pleased that the reviewer appreciated that “*The main finding of the work is interesting, the quality of data is good and results are convincing.*” We thank the reviewer for the helpful suggestions for changes, especially in regards to the scope of the work, which we implemented in the revised paper and address in the point-by-point replies below.

Major

1. **The manuscript is kind of long and tedious, not easy to get points. Should be shorted and tightened. Also, it is need to tighten the rationale and clarify the potential impact of this study.**

Response: We thank the reviewer for suggesting how to improve the clarity and impact of the paper. We agree that the paper was overall long and that some simplification would benefit the presentation of our results. As the reviewer –as well as the other reviewers– points out, the roles of specific layers of the S1 cortex in pain processing remained elusive. To fill this significant knowledge gap required a large number of experiments of different types, thereby making the scope of this study naturally large. In the revision, we did the best we could to make the paper more concise. Accordingly, we streamlined the manuscript to guide the reader to the most significant findings of the study. Specifically, we moved several of the supporting

data into the supplementary material (thermal experiments, L5 stimulation CPA, some details of electrophysiology experiments (pruned and rearranged Figs. 3 and 4), and in favor of new inhibition experiments (e.g. new Fig. 5 and Supplementary Figs. 10, 11), removed more descriptive results and some data entirely (behavioral frequency stimulation data). To make it easier to get the points, we summarize the main findings in the context of the known circuitry in a new graphic in the results (Fig. 4g) and in the discussion (new Fig. 8). We also revised the abstract, introduction and discussion to make the impact more clear. In our opinion, making these changes uncovered a much stronger narrative, which we thank the reviewer for.

2. By only examining hindpaw withdrawal probabilities in response to mechanical stimulation during optical activation of L6-CT neurons, the conclusion that “L6-CT activation induced mechanical hyperalgesia and allodynia in naive animals, and exacerbated these effects in an inflammatory model” is not convincing. Whether the activity of L6-CT neurons was enhanced in the model mice? I am also curious that if optical inhibition of L6-CT neurons induces analgesic effect and conditioned place preference?

Response: We thank the reviewer for raising these points regarding the relationship between L6-CT activity, nociceptive behavior and the inflammatory pain model, which we addressed as detailed below. First, we would like to clarify that all of the original and additional experiments coherently show that L6-CT activation increases sensitivity, both in naive animals and in the inflammatory model. This conclusion is supported by the spontaneous nocifensive behavior (Fig. 1) the von Frey experiments (Fig. 2) – which is a widely accepted proxy for evoked pain – the heat responses (Supplementary Fig. 5), and the aversion test (Fig. 2e-h).

We implemented the three specific points as follows:

- 1) The statement that “L6-CT activation induced mechanical hyperalgesia and allodynia in naive animals, and exacerbated these effects in an inflammatory model” was not intended to imply that CFA injection enhanced L6-CT spiking. We have updated this sentence accordingly: “Taken together, L6-CT activation induced mechanical hyperalgesia and allodynia in naive animals, and furthermore exacerbated hyperalgesia and allodynia in an inflammatory model (Fig. 2b and c).“
- 2) To experimentally address the question whether activity of L6-CT neurons is modulated in the CFA model, we conducted two new experiments. In summary, the electrophysiology and c-Fos data point to small increased firing rates in L6 in the CFA model compared to controls. We describe these experiments and outcomes in detail in our response to comment 5 by reviewer 1 above and kindly refer the reviewer to this section.
- 3) To address the reviewer’s question if optical inhibition of L6-CT neurons induces analgesic effects and conditioned place preference, we did additional conditioning experiments (conditioned place preference, CPP, and conditioned place aversion, CPA) in naive and CFA cohorts and optically inhibited or stimulated L6-CT and L5 neurons.
 - a) Suppressing L6-CT neurons (using the inhibitory opsin stGtACR2, new Supplementary Fig. 10) had only mild or no effects on mechanical or thermal sensitivity, respectively and causes very limited preference that is only

observable in the between-group analysis of PIs (new Supplementary Fig. 11). We also kindly refer the reviewer to our response to reviewer 1's first comment).

- b) In contrast, optogenetic inhibition of L5 increased mechanical sensitivity, and had no effect in the CPP paradigm but instead caused aversion in the CPA paradigm – all of these observations recapitulate the L6-CT stimulation experiments and support the antinociceptive function of L5 neurons that our study suggests (new panels in Figs. 5, 6 and new Supplementary Fig. 11g).
- c) We provide new key data showing that L5-ChR2 stimulation causes a strong place preference in CFA animals, supporting that the antinociceptive effect is not only on acute stimuli (new Fig. 6 h-j).
- d) We compiled a summary of the outcomes of the main behavioral manipulations (old and new) in new Supplementary table 3.

Together, these results provide further insight into the mechanisms of bidirectional modulation of nociception by S1HL output pathways, and in combination with the electrophysiology results, suggest that the pronociceptive function of L6-CT is the result of combined suppression of L5 and enhancement of thalamocortical gain. Thank you for pointing us in this direction, especially the suppression experiments, which produced new key results in the case of L5 suppression.

3. L6-CT neurons are targeted and manipulated using Ntsr1-Cre mouse line, and the authors showed optical activation of L6-CT neurons enhance or suppress neuronal activity of VPL and L5, respectively.

Are these opposite effects caused by direct projections from L6-CT neurons?

If these opposite effects are caused by direct projection from L6-CT neurons, whether these Ntsr1 neurons innervate L5 and VPL by different neurotransmitters?

In addition, L2/3 and L4 neurons showed increased and decreased activity in response to L6-CT neuronal activation. Do Ntsr1 L6-CT neurons release different neurotransmitters in a region specific manner?

Response: We apologize that the L6-CT synaptic properties and functional circuitry was not clearly laid out in the original manuscript and thank the reviewer for pointing this out. We clarified these points in the revised manuscript by explaining the circuitry and also added new summaries (Fig. 4g and Fig. 8) to better anchor our main findings within the context of known circuitry of the L6-CT pathway. We added critical information to the manuscript (lines 628, 639) that L6-CT neurons are glutamatergic (Crandall et al., 2017) and that the inhibition of L5 is indirect via translaminar interneurons in L6 and FS neurons in L5 (Bortone et al., 2014; Frandolig et al., 2019; Kim et al., 2014; Olsen et al., 2012; Puzin & Krieger, 2018). The excitation of VPL is direct via L6-CT synapses onto VPL neurons (Guo et al., 2020). In addition to the including these points at several places in the revised paper (starting at lines 215; 353, 596, 633), we would like to respond to the specific points in relation to our results below:

Are these opposite effects caused by direct projections from L6-CT neurons? If these opposite effects are caused by direct projection from L6-CT neurons, whether these Ntsr1 neurons innervate L5 and VPL by different neurotransmitters?

L6-CT neurons form glutamatergic synapses in the thalamus (Crandall et al., 2015; Guo et al., 2020) and in the cortex, in which they engage excitatory and inhibitory neurons (Bortone et al., 2014; Frandolig et al., 2019; Kim et al., 2014; Olsen et al., 2012). The suppression of L5 is thus indirect via feedforward inhibition, while thalamic excitation is direct. It was shown in

the whisker system that the same neuronal subtype (Ntsr1) as we use to manipulate L6-CT excites the ventroposterior thalamus (VPM) while at the same time suppressing L5 (Pauzin & Krieger, 2018). Thus, we discover here a similar circuit effect in a different cortical area (S1HL) with a novel impact in the context of pain. To clarify these points and highlight the key neuronal circuits involved, we include a new summary figure in the discussion (Fig. 8), and explicitly mention the mechanism at several places (starting at lines 215; 353, 596, 633).

In addition, L2/3 and L4 neurons showed increased and decreased activity in response to L6-CT neuronal activation. Do Ntsr1 L6-CT neurons release different neurotransmitters in a region specific manner?

To the best of our knowledge, it has not been found, and it is indeed unlikely that glutamatergic neurons of the cortex, such as L6-CT neurons, release other fast-acting neurotransmitters (such as GABA). L6-CT neurons suppress other neurons, via multisynaptic feedforward inhibition (Bortone et al., 2014; Frandolig et al., 2019; Olsen et al., 2012; Pauzin & Krieger, 2018), now depicted in Fig. 4g and Fig. 8. The mechanism the reviewer is asking about was demonstrated by (Kim et al., 2014), who showed that L6-CT monosynaptically excite and disynaptically inhibit postsynaptic neurons. They also ruled out monosynaptic release of GABA by L6-CT neurons onto L5 (and L4) neurons. The particular net effect (excitation or feedforward inhibition) onto a particular postsynaptic neuron depends on the connectivity, such that the strength and balance between direct glutamatergic and feedforward inhibitory connections determine the net excitation or suppression. During stimulation of L6-CT *in vivo*, we find excited and inhibited units within the entire L2/3 and L4 populations, but the net effect was excitation for these layers (Fig. 4). To make this more clear, we added a new summary figure of the effects of L6-CT onto all investigated target circuits (new Fig. 4g).

4. For fig.3, “L6-CT activation enhances VPL output to cortex”. The data in the figure showed either L6-CT neuronal activation or mechanical paw stimulation enhance the activity of VPL neurons. How can they conclude VPL outputs to cortex are enhanced without further evidence?

Response: We apologize that this statement was misleading. VPL makes glutamatergic synapses with the TRN, S1 cortex and the posterior insular cortex (see for example (Bokiniec et al., 2022; Guo et al., 2020)). Thus, increased action potential rates in VPL suggest enhanced output to these targets, including the cortex. Our study proposes that this enhanced thalamic throughput – caused by L6-CT stimulation – contributes to the behavioral effects, namely the increased sensory sensitivity, aversion and nociception. This gain mechanism is consistent with L6-CT functions observed in other modalities (see revised introduction and discussion). That L6-CT neurons have been shown to enhance thalamic output was also pointed out by Reviewer 3 “L6 CT cells can influence thalamic throughput and cortical sensitivity”. Thus, we stand by this statement but understand that it could be misunderstood as if we exclusively focussed this study on the effects of this increased thalamic throughput on the cortical recipient neurons. We revised the sentence accordingly to “L6-CT activation enhances VPL spiking output” (line 337).

5. “Both L2/3 and L4 showed robust increased spiking in response to paw stimulation, in both proportions of sensory-encoding units (76% and 61%, for L2/3 and L4, respectively; Fig. 4c) and firing rate modulation (Fig. 4d). In contrast to this relative uniformity of sensory encoding, the effect of L6-CT activation was heterogenous and relatively weak, as some units 25 were enhanced or (suppressed)

by the laser alone (L): L2/3: 31% (46%), L4: 40% (43%).” Since the L6 neurons project to somatosensory thalamus, which reciprocally connects with different layers of somatosensory cortex. It is thus unsurprising that the recorded responses are heterogeneous. Blockade of the projection from the VPL to the S1 is preferred to examine the effect of L6-CT neuronal activation on L2/3 and L4 neurons.

Response: We agree that this particular observation of heterogeneous effects of L6-CT stimulation on individual units in L2/3 and L4 is unsurprising and quite plausible due to the reasons that the reviewer describes. Indeed, L6-CT connections have been proposed to close excitatory thalamocortical loops (Harris & Shepherd, 2015). Consistent with the known intracortical connections of L6-CT neurons onto excitatory and inhibitory cortical neurons (Thomson, 2010), we find both activation and suppression of neurons in L2/3 and L4, with a net activating effect of these layers in response to L6-CT activation (Fig. 4). Conceivably, a blocking experiment of VPL may shift the proportions to less excitation in L4, by removing the recurrent component.

We attempted to do this challenging experiment by recording laser and mechanically-evoked responses across layers in S1 (including L2/3 and L4 as suggested) both before and after injection of muscimol into the VPL (150 nL) to block VPL input into S1. However, we got data insufficient to fully resolve the recurrent component in L2/3 and L4. As expected, mechanical responses in S1 were abolished by blocking VPL, preventing the analysis of the effects of L6-CT on mechanical responses in S1. We discuss this limitation now in the revised discussion: *“Consistent with an increase in thalamic signaling, at the cortical level, L4 and L2/3 responses to mechanical stimuli increased. L6-CT recurrent excitation via the thalamus likely contributes to the net enhancement of L2/3 and L4 (Harris & Shepherd, 2015) and we cannot fully resolve the recurrent component in L2/3 and L4 in the present study. Nevertheless, this enhancement of the VPL – L4 – L2/3 axis suggests that L6-CT can amplify sensory transmission through the canonical circuitry of the lemniscal pathway of the lateral pain system, which carries body signals to S1 (Groh et al., 2017).”*. In our view, fully addressing the recurrent contribution of L6-CT onto cortical dynamics is a fantastic question but is an independent study in itself and the results would not change the main conclusions of our study. There is also the limitation that VPL is essential for the transmission of sensory signals to the S1. A blocking experiment creates a situation in which sensory signaling to S1 is abolished and the modulation of sensory signals by S1 output pathways, which is the focus of our study, cannot be evaluated in this situation. We hope that our response and the clarifications that we made in the manuscript upon this request clearly convey that our conclusions in the manuscript stand strongly without fully resolving the recurrent question here.

6. In the experiments examining the role of L5-PT neurons in pain processing, the conditioned place preference test rather than CPA should be employed to examine whether L5-PT neuronal activation induce preference since previous data showed activation of these neurons decreased mechanical sensitivity in CFA-injected mice. Besides, analyses of the preference indices show that laser stimulation in S1HL in L5-EGFP controls caused comparable avoidance for the laser paired chamber. The authors explained this avoidance may arise entirely from visual avoidance to the laser flashes. However, this avoidance was not observed in L6-EGFP mice which treated with similar light. Therefore, these results need new explanation.

Response: We thank the reviewer for bringing up these two points, which we addressed in the following ways. For the first point, as suggested, we conducted new behavioral

experiments in which we stimulated L5 neurons of mice with a pre-existing pain condition (paw inflammation, CFA model) to test if L5 stimulation causes place preference, consistent with pain relief. We found that L5 stimulation causes a place preference (CPP), while control mice (L5-EGFP) did not show this effect (new Fig. 6h-j). Thus, in full agreement with our original results on L5 stimulation, which decreased mechanical sensitivity, the new CPP results strongly support the antinociceptive function of L5. Furthermore, we show in new experiments a place-aversion phenotype upon L5 suppression (new Fig. 5f, g). Thus, multiple lines of evidence in our study support the antinociceptive role of S1HL L5 neurons. As a result of the reviewer's suggestions, this key finding is now very well supported by several experiments, and we would like to thank the reviewer for motivating these further lines of inquiry

For the second point, we apologize that this was not clear. We can clarify that in contrast to the reviewer's reading, the laser caused a small but measurable avoidance (CPA) also in the L6-EGFP controls (Fig. 2h, see asterisks). In fact, this CPA effect was coherently observed in all control cohorts in this study, including the control cohorts in the new experiments (L6-EGFP: Fig. 2h, Supplementary Fig. 11d; L5-EGFP: Fig. 5g, Supplementary Fig. 9b). We conclude that the laser itself is aversive, but as described in our response to Reviewer 1, we rigorously controlled for this effect. Importantly, Supplementary Fig. 9a shows that while L5 stimulation elicited a greater reduction in the amount of time spent in the paired chamber (relative to the control group), analysis of differences in chamber preference indices (PIs) between experimental and control groups allowed us to isolate the effects of neuronal stimulation from the laser itself (Supplementary Fig. 9b). Here, the PIs of L5-ChR2 and L5-EGFP animals were statistically indifferent (i.e. both, experimental and control animals responded to the laser in the same way, Supplementary Fig. 9b). In contrast, the PIs of L6-ChR2 and L6-EGFP mice were significantly different (i.e. L6-ChR2 mice responded much more strongly to the laser compared to controls, Fig. 2h). This shows that stimulation of L6-CT – but not L5 – causes aversion. To make this more clear, we updated the Results sections and added asterisks and hashtags to all CPA and new conditioned place preference (CPP) analysis to indicate statistical differences between conditions and groups, respectively. We would also kindly ask the reviewer to have a look at our response to comment 6 by reviewer 1, which touches upon the same topic.

7. In vivo recording showed that optical activation of L6 neurons suppress L5 neurons and activate VPL neurons separately. Behaviorally, L6 neuronal activation induced spontaneous nocifensive behavior and L5 neuronal activation normalized inflammatory allodynia. The reciprocal connections between L5, L6 and VPL in pain state is not clear.

Response: Thank you for bringing up this point, which we can clarify. In the new Fig. 8 and Fig. 4g, we summarize the connectivity of the circuitry and the proposed circuit interactions underlying the bidirectional modulation of nociception by S1HL. We revised the manuscript at several places and provide additional summaries and new data on L5 and L6-CT and their interactions (Fig. 4, 5) supporting the circuit mechanisms underlying the observations in the behavior. Specifically, we show in the original and new data that the pronociceptive function of L6-CT is likely the result of combined suppression of L5 and enhancement of thalamocortical gain (via enhancing the VPL-L4-L2/3 axis). We also addressed this question in new experiments and show that optogenetic L5 inhibition enhances L6 spiking (new Fig. 5), in line with a recent study which shows that L5 neurons functionally suppress L6 neurons (Onodera & Kato, 2022). In addition to these revisions please also see our detailed description

of the circuitry in our replies to reviewer 1 (comment 2) and reviewer 3 (comments 1 and 2) who also brought up this point. Additionally, we substantially revised the discussion to accommodate the reviewer's question.

Reviewer #3 (Remarks to the Author):

Ziegler et al. have manipulated cortical activity in the hindlimb primary somatosensory cortex (S1) using cell-type specific mouse driver lines to study the functional impact on the perception of innocuous and noxious sensory signals. Using a combination of optogenetics and in vivo electrophysiology, the authors report that in awake mice, activation of L6 CT cells using the Ntsr1-Cre driver line results in nocifensive behaviors and increased paw sensitivity, as well as exacerbates inflammatory allodynia and elicits an aversive experience. The authors then demonstrate that the pronociceptive effect of L6 CT activation involves enhancing the excitability and sensory throughput of the ventral posterolateral thalamus (VPL) to mechanical paw stimulation in anesthetized mice. In the cortex, the authors found that L6 stimulation completely suppressed L5 activity but had only a modest influence on the overall spiking responses in L4 and L2/3, despite the overall elevated activity in VPL. This observation is largely consistent with the idea that L6 serves as an intracortical gain control (Olsen et al., 2012, Nature). Given L6 stimulation induced both nocifensive behavior and paw hypersensitivity while also eliminating L5 activity, the authors next probed the role of L5 pyramidal tract (PT) neuron activity in nocifensive behaviors and modulating paw sensitivity using the Rbp4-Cre driver line. They report that L5 PT activity in S1 has opposing effects to L6 CT activity, reducing mechanical sensitivity and normalizing inflammatory hypersensitivity.

This study is technically impressive, with some interesting results. The observation that optical stimulation of L6 CT cells can influence thalamic throughput and cortical sensitivity is not novel. Still, as the authors nicely highlight in their manuscript, the role of CT cells in maladaptive hypersensitivity, particularly involving pain perception, has received very little attention. Thus, their finding that L6 CT modulates sensory gain and nociceptive behavior is important to the field. Furthermore, the demonstration that enhancing L5 activity has an opposite effect (or anti-nociceptive function) is also new and potentially interesting. To come to these conclusions, the authors combined in vivo electrophysiological and optical techniques as well as different behavioral approaches in a nicely integrated way. The paper is well written and illustrated, for the most part.

In the manuscript, however, some critical loose ends need to be addressed. Most importantly, the neural circuit mechanism underlying the reported layer-specific bidirectional control of nociceptive behavior.

Response: We thank the reviewer for the insightful and scholarly review of our manuscript and the helpful suggestions. We are delighted about the reviewer's acknowledgment of the importance of the study. The reviewer rightfully points out as a major concern that the link between L6-CT behavioral modulation and L5 suppression represented a critical loose end of the study. As detailed below, we were able to close the missing link, by including new key data showing that direct optogenetic suppression of L5 activity recapitulates the hypersensitivity and aversion phenotype of L6-CT stimulation. This additional data supports our hypothesis

that indeed the L6-CT mediated suppression of L5 mechanistically contributes to the pronociceptive effect of L6-CT stimulation.

My questions and concerns are listed in roughly decreasing order of importance:

1. My main concern has to do with the mechanism underlying the bidirectional modulation of sensory gain and nociceptive behavior. In their first set of experiments, the authors suggest that L6 CT enhancement of nocifensive behaviors and increased paw sensitivity is a consequence of enhanced thalamic throughput and, in parallel, strong suppression of L5 neurons. However, it is unclear how much of the behavioral modulation is due to elevated thalamic activity versus suppression of L5 activity. They sort of address this in their second set of experiments, where they show that elevated L5 activity reduces sensory sensitivity, but again the mechanism underlying how elevated L5 activity decreases sensitivity is unclear (not even investigated). This is critical since the antinociceptive actions of elevated L5 activity are unlikely to involve VPL, at least directly, since L5 cells don't project to VPL.

Response: We thank the reviewer for bringing up the important point about the role of L5 in nociception and its relation to the L6-CT pathway. We addressed this comment in several additional experiments. As suggested, we first investigated how much of the behavioral modulation is due to suppression of L5 activity. We found that direct optogenetic suppression of L5 neurons using the inhibitory opsin stGtACR2 increased mechanical sensitivity and induced aversion (new Fig. 5). Moreover, we more directly investigated if L5 stimulation can ameliorate inflammatory pain in additional real-time place preference tests. Here, mice freely explored two chambers of which the non-preferred chamber was longitudinally paired with optogenetic stimulation of S1HL. L5-ChR2 mice, but not L5-EGFP controls showed real-time conditioned place preference (CPP) for the stimulation-paired chamber (new data in Fig. 6h-j). Thus, L5 stimulation reduces sensitivity in naive animals and causes a place preference in the inflammation model, while suppression of L5 causes hypersensitivity and aversion. In the context of the reviewer's question, the combined results of our L5 and L6-CT investigations suggest that the pronociceptive function of L6-CT (Figs. 1, 2) is the result of simultaneous suppression of L5 (Fig. 4) and enhancement of thalamocortical gain (Figs. 3, 4). We added these new key results at critical places throughout the revised manuscript (lines 75, 420, 648).

Furthermore, the reviewer is correct, that the antinociceptive actions of elevated L5 activity are unlikely to directly involve VPL which does not receive L5 input (Guo et al., 2020). We make this more clear in the revised discussion (line 656). Layer 5 in S1 targets a variety of subcortical circuits along the entire neural axis, such as higher-order thalamic nuclei, superior colliculus, pontine nuclei, basal ganglia, brain stem nuclei and the spinal cord. While we absolutely share the curiosity of the reviewer about the L5 downstream effects (also mentioned in the reviewer's comment 3), in our opinion, it is beyond the scope of the present paper to characterize the putative L5 subcortical targets and recipient cell-types individually. A literature survey showed that L5 projections to the spinal cord and to the periaqueductal gray (PAG) are promising candidates in the context of pain modulation (François et al., 2017; Frezel et al., 2020; Zhang et al., 2015). Indeed, we find that L5 projections from the S1HL target the spinal cord and PAG (preliminary data). An interesting hypothesis for future studies is that the L5 pathway from S1HL can reduce ascending sensory signals and/or activate descending modulation via these targets, for example via the S1 L5 projections to the dorsal horn (Frezel et al., 2020).

In addition to including the new data described above, we significantly revised the discussion and added a summary (Fig. 8) to account for these important points brought up by the reviewer.

2. Related to point 1 above, it would be interesting to know more about the bi-directional gain control of each layer in isolation, as well as how L5 and L6 interact. Specifically, what is the impact of L5 activation/suppression on L6 activity? Does L5 suppression alone cause hypersensitivity? What happens when L6 activity is suppressed? Does sensitivity decrease? Does L5 activity increase?

Response: Thank you for bringing up these interesting questions which we addressed in additional experiments and by discussing previous work as detailed below. We addressed the points in order and the paper is much better as a result of the data collected from these experiments. We specifically addressed the impact of suppression of L5 on behavior and on L6 activity; we further suppressed L6 activity and observed mild effects in the behavior and see that in electrophysiology experiments, under both anesthetized and awake experimental conditions, L6-CT activity is so low that it cannot be further suppressed. We also added a new discussion paragraph about the interactions of L5 and L6 (starting at line 673).

Specifically, what is the impact of L5 activation/suppression on L6 activity?

Response: In regards to the effect of L5 activation and suppression on L6 activity, a recent study by (Onodera & Kato, 2022) provides an excellent quantitative and exhaustive description of the interactions of L5 neurons with the cortical column in the primary auditory cortex. Using a very similar optogenetic approach to stimulate the L5 subtype (Rbp4-Cre) as we used in our study, the authors found that in response to L5-ChR2 stimulation, most of the L6 (63%) units were suppressed. In turn, L5 suppression resulted in an overall activation of L6 units (52% were activated). The effects of L5 manipulations may differ between cortical areas. However, in new experiments motivated by this comment, we were able to recapitulate some of the findings of (Onodera & Kato, 2022) in the S1HL cortex. We found that suppression of L5 using the inhibitory opsin stGtACR2 leads to a net enhancement of L6 in S1HL in awake animals (42% were activated vs 10% suppressed) (new Fig. 5), consistent with disinhibition of L6. Together, these results suggest that L5 has a net suppressive effect on L6 and L6 has a suppressive effect on L5 (Fig. 4). Hence, the two cortical output pathways appear to mutually suppress each other (please see paragraph starting at line 673). The suggested suppression of L5 gave rise to new key insights and made the revised paper much better. We therefore summarize these experiments in a new main figure (Fig. 5) and thank the reviewer for pointing us in this direction.

Does L5 suppression alone cause hypersensitivity?

We tested potential hypersensitivity upon L5 suppression in a new set of experiments. L5 neurons in S1HL were optogenetically suppressed using the stGtACR2 inhibitory opsin (Mahn et al., 2018). We observed that L5 suppression increased mechanical sensitivity in these mice (Fig. 5e) and induced a place aversion phenotype (Fig. 5f, g). We thank the reviewer for stimulating these additional experiments, as these revealed the important finding that L5 inhibition in the S1HL cortex recapitulates pronociceptive effects of L6-CT activation. Thus, the pronociceptive effect of L6-CT activation seems to indeed work (at least in part) via suppression of the antinociceptive L5 pathway. This is a new key data point in the revised paper.

What happens when L6 activity is suppressed? Does sensitivity decrease? Does L5 activity increase?

To test for these effects we optogenetically suppressed L6-CT in S1HL using the stGtACR2 inhibitory opsin (Mahn et al. 2018). We observed no effect on heat sensitivity and a slight increase in mechanical sensitivity and very limited preference that is only observable in the between-group analysis of PIs (Supplementary Fig. 11). For the second question whether L5 activity is increased upon suppression of L6-CT, we encountered a caveat in experimentally suppressing L6-CT activity. We kindly refer the reviewer also to our response to reviewer 1, last point in comment 1. In brief, L6-CT activity was low under our experimental conditions and could not efficiently be suppressed further, even though the opsin works very efficiently as we show *in vitro* (Supplementary Fig. 10). How can our observation that L6-CT activation causes hypersensitivity be reconciled with the average low activity of L6-CT neurons under the experimental conditions that have been used in our and in previous studies? We interpret the reported average low L6-CT activity as a result of experimental constraints (anesthesia, head-fixed recordings), rather than a biological feature of L6, although it may also be that L6 neurons are only very transiently and sparsely activated under particular behavioral or state-dependent conditions which have yet to be identified. Indeed L6-CT was referred to as “mysterious creatures of the deep” whose activity may be controlled by high-order cortical areas (Harris & Shepherd, 2015) which have yet to be discovered. The conclusion that L6-CT activation causes hypersensitivity is still valid, even though it is currently not known what drives L6-CT neurons under physiological or pathological conditions. We discuss these important points now starting at line 662. Importantly, our observation that L6-CT activity can cause adverse sensory experiences points in the direction that the activity of L6-CT neurons must be tightly regulated and that high sensory gain is not necessarily beneficial.

We fully agree with the reviewer that further electrophysiological dissection of L5 - L6 interactions would be interesting. We therefore added a new discussion paragraph (starting at line 673). We do however also feel that our study already provides a number of novel landmark insights into the involvement of S1 in pain, and importantly the distinct functions of underlying S1 output pathways. All our conclusions are stringently supported by extensive data and our conclusions hold up in the extensive additional data collected during the revision. Since the scope of the paper is already quite large, we think that further investigations about L5 and L6 interactions would much better fit into a separate study. We hope that our response and the clarifications that we made in the manuscript upon this request clearly convey that our conclusions in the manuscript stand strongly and would not fundamentally change by fully resolving the effect of L6-CT suppression on L5 in this present study.

3. The authors also don't consider another feature of the circuit: the higher-order posterior medial thalamus (POm). They present data about L6 modulatory inputs to VPL, but never report on the influence on the POm. This seems critical since this thalamic region is thought to serve higher-order functions, and it receives input directly from L5 and L6 cells.

Response: The reviewer is correct, POm receives input from L5 and L6-CT neurons in S1HL (Guo et al., 2020). We addressed this experimentally and found that mechanical responses in POm were not comparable to the robust responses in VPL (new Supplementary Fig. 12). However, POm spontaneous activity was affected by L6-CT stimulation albeit more weakly and heterogeneously, with lower median modulation indices. In contrast, VPL was on average strongly enhanced by L6-CT stimulation (Fig. 3 and direct comparison Supplementary Fig.

12). We interpret that the heterogeneity in POM arises from direct L6-CT connections, which are excitatory, and indirectly via the L6-CT mediated suppression of L5 in S1HL (Fig. 4) which is a main driver of POM activity (Guo et al., 2020; Mease et al., 2016). In summary, we can confirm that L6-CT input also modulates POM, which may contribute to the pronociceptive phenotype following L6-CT stimulation. In addition to the new POM data in Supplementary Fig. 12, we briefly discuss POM in the results and discussion (lines 314; 606).

4. Throughout the manuscript, we are led to believe that the Rbp4-Cre line is specific to L5 PT cells, but we are provided no evidence that this is the case for S1 HL. Based on the original report and subsequent papers (as well as the mouse connectivity map from allen institute), I believe this mouse line expresses cre in both PT and IT cells (Gerfen et al., 2013, Neuron). Without such new data, the author's conclusions about L5 PT cell involvement must be revised.

Response: Thank you for pointing this out and we apologize for this oversight. In a recent paper about the S1HL the authors refer to L5-PT using this mouse line (Guo et al., 2020) but the reviewer is absolutely correct that Rbp4-neurons in S1HL likely express Cre in both PT and IT L5 neurons. Accordingly, we replaced the L5-PT terminology in the paper and instead refer to "L5".

5. Additional information regarding the delay between light stimulation and behavior would be helpful for Figures 1 and 5.

Response: Behavioral reactions were considered within the 5 s laser stimulation window (now added to the captions and methods section). Hence, the delays in Figs. 1 and 5 that the reviewer is referring to are < 5 s. Please see also supplementary videos of these behaviors, which show that the L6-CT evoked paw lifting was tightly coupled to the laser stimulation. In Figs. 2, 5, 6 we also show mechanically-evoked behavior. The mechanical stimulus was applied within 1 s after the onset of the 5 s laser stimulation. Here, again behavioral reactions were considered within the 5 s laser stimulation window. We give a full description in the methods "von Frey test". More precise measurements of the delay between mechanical stimulation and behavioral reactions have been worked out in a previous study using the same von Frey paradigm and were approximately within 200-600 ms (Tan et al., 2019).

6. Identification of opsin expressing cells should be described in more detail in Figure 1 and 5. Plots showing the criteria used to distinguish optotagged L5/L6 cells from non-optotagged cells and how FS-like neurons were separated from optotagged cells.

Response: We followed this suggestion and added a description of our optotagging strategy and additional data in new Supplementary Fig. 2 and in the methods "Classification of putative cell types". Briefly, the tagging procedure entailed giving 10 Hz laser trains (10 ms pulse length) to either Rbp4-Cre-ChR2-EYFP (n = 2) or Ntsr1-Cre-ChR2-EYFP mice (n = 3) as part of a 5 second on, 5 second off protocol (> 1000 pulses in total per mouse). We then calculated the mean first-spike latency and the standard deviation of the first spike latency to each 10 ms laser pulse for each single unit from a pooled dataset of each mouse line. Putative FS units below (delta trough-to-second-peak of extracellular mean waveform < 215 us; Schmitt et al., 2017) were removed from the tagged population. We used first spike latency and standard deviation to separate optotagged units from non optotagged units and found the best separation with mean first spike latency < 9.5 ms and standard deviation of < 2 ms and < 3.5 ms for L6-CT units (83/384) and L5 units (57/274), respectively. The median depth of the

tagged L5 units was $-665.5 \mu\text{m}$ (interquartile range = $151.5 \mu\text{m}$), whereas the median depth of the tagged L6-CT units was $-1142 \mu\text{m}$ (interquartile range = $150.75 \mu\text{m}$), in line with our histological findings and previous literature in the S1 (O'Connor et al., 2010). We performed this separation only on optotagged populations to verify cell-type-specific control of the target cells; for the main analyses of optical stimulation and inhibition on cortical layers, we included all units that fulfilled quality metrics (now better described in the methods, page 32 "Classification of putative cell types")

Minor

1. **Page 4, line 1: should be "and their corticothalamic axons/terminals".**

Response: Thank you, we changed the text.

2. **Figure 1d: the values on the y-axis are small and hard to see.**

Response: Thank you, we improved the readability.

3. **The viral titer and expression time should be reported**

Response: Thank you, we added missing information to the methods.

4. **change Hargreave's to Hargreaves (e.g. Fig 2)**

Response: Thank you, done throughout.

References

- Beltramo, R., D'Urso, G., Dal Maschio, M., Farisello, P., Bovetti, S., Clovis, Y., Lassi, G., Tucci, V., De Pietri Tonelli, D., & Fellin, T. (2013). Layer-specific excitatory circuits differentially control recurrent network dynamics in the neocortex. *Nature Neuroscience*, *16*(2), 227–234. <https://doi.org/10.1038/nn.3306>
- Bokiniec, P., Whitmire, C. J., Leva, T. M., & Poulet, J. F. A. (2022). Brain-wide connectivity map of mouse thermosensory cortices. *Cerebral Cortex*. <https://doi.org/10.1093/cercor/bhac386>
- Bortone, D. S., Olsen, S. R., & Scanziani, M. (2014). Translaminar inhibitory cells recruited by layer 6 corticothalamic neurons suppress visual cortex. *Neuron*, *82*(2), 474–485. <https://doi.org/10.1016/j.neuron.2014.02.021>
- Chang, Y., Yan, L.-H., Zhang, F.-K., Gong, K.-R., Liu, M.-G., Xiao, Y., Xie, F., Fu, H., & Chen, J. (2008). Spatiotemporal characteristics of pain-associated neuronal activities in primary somatosensory cortex induced by peripheral persistent nociception.

- Neuroscience Letters*, 448(1), 134–138. <https://doi.org/10.1016/j.neulet.2008.08.090>
- Clayton, K. K., Williamson, R. S., Hancock, K. E., Tasaka, G.-I., Mizrahi, A., Hackett, T. A., & Polley, D. B. (2021). Auditory Corticothalamic Neurons Are Recruited by Motor Preparatory Inputs. *Current Biology: CB*, 31(2), 310–321.e5. <https://doi.org/10.1016/j.cub.2020.10.027>
- Crandall, S. R., Cruikshank, S. J., & Connors, B. W. (2015). A Corticothalamic Switch: Controlling the Thalamus with Dynamic Synapses. *Neuron*, 86(3), 768–782. <https://doi.org/10.1016/j.neuron.2015.03.040>
- Crandall, S. R., Patrick, S. L., Cruikshank, S. J., & Connors, B. W. (2017). Infrabarrels Are Layer 6 Circuit Modules in the Barrel Cortex that Link Long-Range Inputs and Outputs. *Cell Reports*, 21(11), 3065–3078. <https://doi.org/10.1016/j.celrep.2017.11.049>
- de Kock, C. P. J., & Sakmann, B. (2009). Spiking in primary somatosensory cortex during natural whisking in awake head-restrained rats is cell-type specific. *Proceedings of the National Academy of Sciences of the United States of America*, 106(38), 16446–16450. <https://doi.org/10.1073/pnas.0904143106>
- François, A., Low, S. A., Sypek, E. I., Christensen, A. J., Sotoudeh, C., Beier, K. T., Ramakrishnan, C., Ritola, K. D., Sharif-Naeini, R., Deisseroth, K., Delp, S. L., Malenka, R. C., Luo, L., Hantman, A. W., & Scherrer, G. (2017). A Brainstem-Spinal Cord Inhibitory Circuit for Mechanical Pain Modulation by GABA and Enkephalins. *Neuron*, 93(4), 822–839.e6. <https://doi.org/10.1016/j.neuron.2017.01.008>
- Frändolig, J. E., Matney, C. J., Lee, K., Kim, J., Chevée, M., Kim, S.-J., Bickert, A. A., & Brown, S. P. (2019). The Synaptic Organization of Layer 6 Circuits Reveals Inhibition as a Major Output of a Neocortical Sublamina. *Cell Reports*, 28(12), 3131–3143.e5. <https://doi.org/10.1016/j.celrep.2019.08.048>
- Frezel, N., Platonova, E., Voigt, F. F., Mateos, J. M., Kastli, R., Ziegler, U., Karayannis, T., Helmchen, F., Wildner, H., & Zeilhofer, H. U. (2020). In-Depth Characterization of Layer 5 Output Neurons of the Primary Somatosensory Cortex Innervating the Mouse Dorsal Spinal Cord. *Cerebral Cortex Communications*, 1(1).

<https://doi.org/10.1093/texcom/tgaa052>

Gerfen, C. R., Paletzki, R., & Heintz, N. (2013). GENSAT BAC cre-recombinase driver lines to study the functional organization of cerebral cortical and basal ganglia circuits.

Neuron, 80(6), 1368–1383. <https://doi.org/10.1016/j.neuron.2013.10.016>

Gong, S., Doughty, M., Harbaugh, C. R., Cummins, A., Hatten, M. E., Heintz, N., & Gerfen, C. R. (2007). Targeting Cre recombinase to specific neuron populations with bacterial artificial chromosome constructs. *The Journal of Neuroscience: The Official Journal of the Society for Neuroscience*, 27(37), 9817–9823.

<https://doi.org/10.1523/JNEUROSCI.2707-07.2007>

Groh, A., Krieger, P., Mease, R. A., & Henderson, L. (2017). Acute and chronic pain processing in the thalamocortical system of humans and animal models. *Neuroscience*.

<https://doi.org/10.1016/j.neuroscience.2017.09.042>

Guo, K., Yamawaki, N., Barrett, J. M., Tapias, M., & Shepherd, G. M. G. (2020). Cortico-Thalamo-Cortical Circuits of Mouse Forelimb S1 Are Organized Primarily as Recurrent Loops. *The Journal of Neuroscience: The Official Journal of the Society for Neuroscience*, 40(14), 2849–2858.

<https://doi.org/10.1523/JNEUROSCI.2277-19.2020>

Harris, K. D., & Shepherd, G. M. G. (2015). The neocortical circuit: themes and variations.

Nature Neuroscience, 18(2), 170–181. <https://doi.org/10.1038/nn.3917>

Kim, J., Matney, C. J., Blankenship, A., Hestrin, S., & Brown, S. P. (2014). Layer 6

Corticothalamic Neurons Activate a Cortical Output Layer, Layer 5a. *Journal of Neuroscience*, 34(29), 9656–9664.

<https://doi.org/10.1523/JNEUROSCI.1325-14.2014>

Kirchgessner, M. A., Franklin, A. D., & Callaway, E. M. (2021). Distinct “driving” versus

“modulatory” influences of different visual corticothalamic pathways. *Current Biology: CB*, 31(23), 5121–5137.e7.

<https://doi.org/10.1016/j.cub.2021.09.025>

Mahn, M., Gibor, L., Patil, P., Cohen-Kashi Malina, K., Oring, S., Printz, Y., Levy, R., Lampl, I., & Yizhar, O. (2018). High-efficiency optogenetic silencing with soma-targeted anion-conducting channelrhodopsins. *Nature Communications*, 9(1), 4125.

<https://doi.org/10.1038/s41467-018-06511-8>

- Mease, R. A., Krieger, P., & Groh, A. (2014). Cortical control of adaptation and sensory relay mode in the thalamus. *Proceedings of the National Academy of Sciences of the United States of America*, *111*(18), 6798–6803. <https://doi.org/10.1073/pnas.1318665111>
- Mease, R. A., Sumser, A., Sakmann, B., & Groh, A. (2016). Corticothalamic Spike Transfer via the L5B-POm Pathway in vivo. *Cerebral Cortex*, *26*(8), 3461–3475. <https://doi.org/10.1093/cercor/bhw123>
- O'Connor, D. H., Peron, S. P., Huber, D., & Svoboda, K. (2010). Neural activity in barrel cortex underlying vibrissa-based object localization in mice. *Neuron*, *67*(6), 1048–1061. <https://doi.org/10.1016/j.neuron.2010.08.026>
- Olsen, S. R., Bortone, D. S., Adesnik, H., & Scanziani, M. (2012). Gain control by layer six in cortical circuits of vision. *Nature*, *483*(7387), 47–52. <https://doi.org/10.1038/nature10835>
- Onodera, K., & Kato, H. K. (2022). Translaminar recurrence from layer 5 suppresses superficial cortical layers. *Nature Communications*, *13*(1), 2585. <https://doi.org/10.1038/s41467-022-30349-w>
- Pauzin, F. P., & Krieger, P. (2018). A Corticothalamic Circuit for Refining Tactile Encoding. *Cell Reports*, *23*(5), 1314–1325. <https://doi.org/10.1016/j.celrep.2018.03.128>
- Prasad, J. A., Carroll, B. J., & Sherman, S. M. (2020). Layer 5 corticofugal projections from diverse cortical areas: variations on a pattern of thalamic and extra-thalamic targets. *The Journal of Neuroscience: The Official Journal of the Society for Neuroscience*. <https://doi.org/10.1523/JNEUROSCI.0529-20.2020>
- Qi, J., Ye, C., Naskar, S., Inácio, A. R., & Lee, S. (2022). Posteromedial thalamic nucleus activity significantly contributes to perceptual discrimination. *PLoS Biology*, *20*(11), e3001896. <https://doi.org/10.1371/journal.pbio.3001896>
- Quiton, R. L., Masri, R., Thompson, S. M., & Keller, A. (2010). Abnormal activity of primary somatosensory cortex in central pain syndrome. *Journal of Neurophysiology*, *104*(3), 1717–1725. <https://doi.org/10.1152/jn.00161.2010>
- Spacek, M. A., Crombie, D., Bauer, Y., Born, G., Liu, X., Katzner, S., & Busse, L. (2022). Robust effects of corticothalamic feedback and behavioral state on movie responses in

mouse dLGN. *eLife*, 11. <https://doi.org/10.7554/eLife.70469>

Tan, L. L., Oswald, M. J., Heini, C., Retana Romero, O. A., Kaushalya, S. K., Monyer, H., & Kuner, R. (2019). Gamma oscillations in somatosensory cortex recruit prefrontal and descending serotonergic pathways in aversion and nociception. *Nature Communications*, 10(1), 983. <https://doi.org/10.1038/s41467-019-08873-z>

Tan, L. L., Pelzer, P., Heini, C., Tang, W., Gangadharan, V., Flor, H., Sprengel, R., Kuner, T., & Kuner, R. (2017). A pathway from midcingulate cortex to posterior insula gates nociceptive hypersensitivity. *Nature Neuroscience*, 20(11), 1591–1601. <https://doi.org/10.1038/nn.4645>

Thomson, A. M. (2010). Neocortical layer 6, a review. *Frontiers in Neuroanatomy*, 4, 13. <https://doi.org/10.3389/fnana.2010.00013>

Vélez-Fort, M., Rousseau, C. V., Niedworok, C. J., Wickersham, I. R., Rancz, E. A., Brown, A. P. Y., Strom, M., & Margrie, T. W. (2014). The stimulus selectivity and connectivity of layer six principal cells reveals cortical microcircuits underlying visual processing. *Neuron*, 83(6), 1431–1443. <https://doi.org/10.1016/j.neuron.2014.08.001>

Zhang, Y., Zhao, S., Rodriguez, E., Takatoh, J., Han, B.-X., Zhou, X., & Wang, F. (2015). Identifying local and descending inputs for primary sensory neurons. *The Journal of Clinical Investigation*, 125(10), 3782–3794. <https://doi.org/10.1172/JCI81156>

REVIEWERS' COMMENTS

Reviewer #1 (Remarks to the Author):

The authors answered all of our comments. Good job!!

Reviewer #2 (Remarks to the Author):

The authors have addressed largely my comments. I support publication of this work in light of its originality and the impact may represent for the field.

Reviewer #3 (Remarks to the Author):

The authors have addressed all of my previous concerns. In addition, they have performed several additional experiments that strengthen this study. The revised manuscript is well-written, and I enjoyed reading it. Finally, I want to congratulate the authors.

minor

Page 6, Line 154: "Supplementary Figs 2, 5." – should this be Figures 3, 5?

Figure 6d: It should be noted in the legend if the L6CT data is the same as plotted in Figure 1F.

REVIEWERS' COMMENTS to Revision

Response: We thank the reviewers for their advice to publish our study.

Reviewer #1 (Remarks to the Author):

The authors answered all of our comments. Good job!!

Reviewer #2 (Remarks to the Author):

The authors have addressed largely my comments. I support publication of this work in light of its originality and the impact may represent for the field.

Reviewer #3 (Remarks to the Author):

The authors have addressed all of my previous concerns. In addition, they have performed several additional experiments that strengthen this study. The revised manuscript is well-written, and I enjoyed reading it. Finally, I want to congratulate the authors.

minor

Page 6, Line 154: "Supplementary Figs 2, 5." – should this be Figures 3, 5?

Figure 6d: It should be noted in the legend if the L6CT data is the same as plotted in Figure 1F.

Response: Thank you, we implemented these points in the final revision.